# ResKoopNet: Learning Koopman Representations for Complex Dynamics with Spectral Residuals

Yuanchao Xu [* 1]    Kaidi Shao [* 2]    Nikos Logothetis [2]    Zhongwei Shen [1]

## Abstract

Analyzing the long-term behavior of high-dimensional nonlinear dynamical systems remains a significant challenge. While the Koopman operator framework provides a powerful global linearization tool, current methods for approximating its spectral components often face theoretical limitations and depend on predefined dictionaries. Residual Dynamic Mode Decomposition (ResDMD) advanced the field by introducing the *spectral residual* to assess Koopman operator approximation accuracy; however, its approach of only filtering precomputed spectra prevents the discovery of the operator's complete spectral information, a limitation known as the 'spectral inclusion' problem. We introduce ResKoopNet (Residual-based Koopman-learning Network), a novel method that directly addresses this by explicitly minimizing the *spectral residual* to compute Koopman eigenpairs. This enables the identification of a more precise and complete Koopman operator spectrum. Using neural networks, our approach provides theoretical guarantees while maintaining computational adaptability. Experiments on a variety of physical and biological systems show that ResKoopNet achieves more accurate spectral approximations than existing methods, particularly for high-dimensional systems and those with continuous spectra, which demonstrates its effectiveness as a tool for analyzing complex dynamical systems.

## 1. Introduction

In the analysis of complex dynamical systems, a fundamental challenge is accurately capturing and understanding the long-term behavior of highly nonlinear systems, particularly those with high dimensionality. Various data-driven methods (Brunton & Kutz, 2019; Schetzen, 2006; Wiggins, 2003; Slotine & Li, 1991; Tu et al., 2014; Lan & Mezić, 2013; Mezić, 2005; Xu et al., 2025; Ishikawa et al., 2024) have been developed to address this challenge. Among these approaches, the Koopman operator framework (Koopman, 1931; Koopman & Neumann, 1932) has emerged as a powerful tool due to its ability to globally linearize nonlinear systems. Unlike local linearization methods (Hartman, 1960; Grobman, 1959), which only approximate dynamics near fixed points, the Koopman operator transforms the entire system into a linear form within an infinite-dimensional function space, enabling the application of spectral analysis techniques to study complex dynamics.

Despite its promise, practical computational challenges arise from the infinite-dimensional nature of the Koopman operator. Numerical methods such as Extended Dynamic Mode Decomposition (EDMD) have been developed to approximate the Koopman operator using a finite set of observables, which makes it possible to extract dynamic modes from data. Important convergence results for EDMD were established by Korda & Mezić (2018), which proved that as the number of samples increases, EDMD converges to the $L^2$-orthogonal projection of the Koopman operator on a finite-dimensional subspace. They further showed convergence in the strong operator topology as the dimension of the subspace increases, and that accumulation points of the spectra of EDMD correspond to eigenvalues of the Koopman operator with associated eigenfunctions converging weakly, provided the weak limit is nonzero. However, despite these theoretical advances, EDMD faces notable limitation in spectral approximation. It suffers from spectral pollution, where discretization introduces spurious eigenvalues unrelated to the true operator. Moreover, EDMD struggles to accurately capture continuous spectra, which are crucial for characterizing chaotic and complex systems. As a result, EDMD can miss critical information about the underlying dynamics, particularly in systems exhibiting rich, complex

---

[*]Equal contribution  [1]Department of Mathematical and Statistical Science, University of Alberta, Edmonton, Canada [2]International Center for Primate Brain Research, Chinese Academy of Sciences, Shanghai, China. Correspondence to: Zhongwei Shen <zhongwei@ualberta.ca>.

*Proceedings of the 42nd International Conference on Machine Learning*, Vancouver, Canada. PMLR 267, 2025. Copyright 2025 by the author(s).

behavior.

To address these limitations, Residual Dynamic Mode Decomposition (ResDMD) (Colbrook & Townsend, 2024) was introduced to offer convergence guarantees through a *spectral residual* that measures how well the estimated Koopman spectra are. By assessing convergence of spectral residual, ResDMD eliminates spurious spectral components that do not represent the system's true dynamics, which enhances the reliability of spectral estimation. ResDMD also introduces the concept of pseudospectrum, regions where the spectral content is significant even without discrete eigenvalues, which is particularly valuable for analyzing systems with continuous spectra. However, ResDMD only filters precomputed polluted spectra, which limits its ability to independently refine spectral estimates.

In this paper, we propose Residual-based Koopman-learning Network (ResKoopNet), which overcomes these limitations by minimizing spectral residuals over eigenpairs while exploring the optimal dictionary space. ResKoopNet employs neural networks to optimize dictionary functions for the Koopman invariant subspace, which enables both discrete eigenvalue computation and pseudospectrum analysis. Through experiments on both simulation models and high-dimensional real-world systems, we demonstrate its superior accuracy and scalability over other methods.

## 2. Preliminary on Koopman Operator

Consider a discrete-time dynamical system $(\Omega, \mu)$ governed by a map $F : \Omega \to \Omega$, where $\Omega \subseteq \mathbb{R}^d$ is the state space, and $\mu$ is a probability measure. The evolution of the system is described by:

$$x_{k+1} = F(x_k), \quad k \in \mathbb{Z}^+.$$

The Koopman operator $\mathcal{K}$ acts on observable $g \in \mathcal{F}$ as:

$$\mathcal{K}g = g \circ F,$$

where the inner product and corresponding norm for $\mathcal{F}$ is defined as $\langle f, g \rangle_\mu := \int_\Omega \bar{f} g \, d\mu$ and $\|f\|_\mu := \sqrt{\langle f, f \rangle_\mu}$, respectively. Notice that $\mathcal{K}$ is linear even though $F$ is nonlinear. Another key aspect of modern Koopman operator theory is Koopman Mode Decomposition (KMD) (Mezić, 2005), which represents system dynamics through its spectral components. In particular, for a given observable $g$, KMD provides the expansion as following

$$g(x_n) = [\mathcal{K}^n g](x_0)$$
$$= \underbrace{\sum_{\lambda \in \sigma_p(\mathcal{K})} c_\lambda \lambda^n \varphi_\lambda(x_0)}_{\text{discrete spectrum}} + \underbrace{\int_{[-\pi, \pi]_{per}} e^{in\theta} \phi_{\theta, g}(x_0) \, d\theta}_{\text{continuous spectrum}},$$

where $\varphi_\lambda$ are the eigenfunctions of $\mathcal{K}$, $c_\lambda$ are expansion coefficients, and $\phi_{\theta, g}$ represents the contribution from the

continuous spectrum. $[-\pi, \pi]_{per}$ denotes a periodic interval. The discrete spectrum is particularly important for insights into long-term behavior, such as periodicity and stability, while the continuous spectrum characterizes complex behaviors such as mixing, chaos, and transport phenomena (Arbabi & Mezic, 2017; Yang et al., 2022; Brunton et al., 2017). Our analysis emphasizes finding these spectral components including both discrete and continuous part from KMD.

Extended Dynamic Mode Decomposition (EDMD) (Williams et al., 2015) is a widely used data-driven method for approximating the Koopman operator and its spectral components in KMD. It selects a set of observables $\{\psi_i\}_{i=1}^{N_K}$ to form a subspace $V_{N_K} = \text{span}\{\psi_i\}_{i=1}^{N_K}$. The method constructs a finite-dimensional approximation of the Koopman operator by solving a least-square problem based on system snapshots and selected dictionary, allowing the computation of eigenvalues, eigenfunctions, and Koopman modes. While common dictionaries include polynomials, Fourier basis, and RBF functions, selecting an optimal dictionary remains system-dependent and challenging. Given data snapshots $\{(x_i, y_i)\}_{i=1}^m$ with $y_i = F(x_i)$ and $\{x_i\}_{i=1}^m$ i.i.d. by $\mu$, two data matrices $\Psi_X$ and $\Psi_Y$ are formed by evaluating the dictionary on these data points:

$$\Psi_X := \begin{bmatrix} \psi_1(x_1) & \dots & \psi_{N_K}(x_1) \\ \vdots & \ddots & \vdots \\ \psi_1(x_m) & \dots & \psi_{N_K}(x_m) \end{bmatrix},$$

$$\Psi_Y := \begin{bmatrix} \psi_1(y_1) & \dots & \psi_{N_K}(y_1) \\ \vdots & \ddots & \vdots \\ \psi_1(y_m) & \dots & \psi_{N_K}(y_m) \end{bmatrix}.$$

EDMD computes the approximated Koopman matrix representation by

$$K = \Psi_X^\dagger \Psi_Y,$$

where $\Psi_X^\dagger$ is the pseudoinverse of $\Psi_X$. The eigenvalues of $K$ provide approximation of the Koopman operator's discrete spectrum, and the Koopman eigenfunctions $\phi_i$ are approximated as $\phi_i = \mathbf{\Psi} \mathbf{v}_i$, where $\mathbf{v}_i \in \mathbb{C}^{N_K}$ is the $i$-th eigenvector of $K$ and

$$\mathbf{\Psi} := [\psi_1, \dots, \psi_{N_K}]$$

is a (row) vector of observables from the dictionary.

## 3. Koopman Operator Learning

EDMD directly approximates the Koopman operator and its spectral components from data but suffers from spectral pollution, where discretization of the infinite-dimensional operator introduces spurious eigenvalues that have no relation

to the true spectrum. Residual Dynamic Mode Decomposition (ResDMD) (Colbrook & Townsend, 2024) improves spectral accuracy by filtering out these polluted eigenvalues via *spectral residual* measurement. However, it only filters precomputed spectra and cannot fully discover the Koopman operator's complete spectral information, a limitation known as the 'spectral inclusion' problem where parts of the true spectrum may be missed entirely.

To address these issues, we propose Residual-based Koopman-learning Network (ResKoopNet), which minimizes *spectral residual* over eigenpairs while exploring the optimal dictionary space. This approach identifies a more precise and complete spectrum of the Koopman operator while ensuring all learned dictionary functions satisfy small *spectral residual* criteria. We note that while we employ a simple neural network as our optimizer in this work, the specific choice of optimization tool can be adapted based on the data structure and properties of the dynamical system being studied.

### 3.1. ResDMD Review

Now, suppose we have an estimated eigenvalue-eigenfunction pair $(\lambda, \phi)$ of $\mathcal{K}$ where $\lambda \in \mathbb{C}$ and $\phi = \mathbf{\Psi v} = \sum_{i=1}^{N_K} \psi_i v_i \in V_{N_K}$ is expressed in the dictionary functions. Assuming $\phi$ is normalized, i.e., $\|\phi\|_\mu = 1$, the *spectral residual* that measures the accuracy of this eigenpair is defined as:

$$
\begin{aligned}
res(\lambda, \phi)^2 &:= \frac{\int_\Omega |\mathcal{K}\phi(x) - \lambda\phi(x)|^2 \, d\mu(x)}{\int_\Omega |\phi(x)|^2 \, d\mu(x)} \\
&= \sum_{i,j=1}^{N_K} \bar{v}_i [\langle \mathcal{K}\psi_i, \mathcal{K}\psi_j \rangle_\mu - \lambda \langle \psi_i, \mathcal{K}\psi_j \rangle_\mu \\
&\quad - \bar{\lambda} \langle \mathcal{K}\psi_i, \psi_j \rangle_\mu + |\lambda|^2 \langle \psi_i, \psi_j \rangle_\mu] v_j,
\end{aligned}
\tag{1}
$$

where $\bar{v}_i, \bar{\lambda}$ denote the complex conjugates of $v_i, \lambda$.

This *spectral residual* in Eq. (1) quantifies how far the estimated eigenpair deviates from the true spectrum. In practice, we can approximate this residual using data samples through Galerkin methods (Boyd, 2013; Colbrook & Townsend, 2024); more specifically, the following convergence results hold as the number of data points $m \to \infty$:

$$
\begin{aligned}
\lim_{m \to \infty} \frac{1}{m} [\Psi_X^* \Psi_X]_{ij} &= \langle \psi_i, \psi_j \rangle_\mu, \\
\lim_{m \to \infty} \frac{1}{m} [\Psi_X^* \Psi_Y]_{ij} &= \langle \psi_i, \mathcal{K}\psi_j \rangle_\mu, \\
\lim_{m \to \infty} \frac{1}{m} [\Psi_Y^* \Psi_Y]_{ij} &= \langle \psi_i, \mathcal{K}^*\mathcal{K}\psi_j \rangle_\mu,
\end{aligned}
\tag{2}
$$

where $*$ denotes conjugate transpose of matrix or adjoint operator. Using these relationships, the *spectral residual* in

Eq. (1) can be approximated as (see A.2 for more details):

$$
\begin{aligned}
\widehat{res}(\lambda, \phi)^2 &:= \frac{1}{m}\mathbf{v}^*[\Psi_Y^* \Psi_Y - \lambda(\Psi_X^* \Psi_Y)^* \\
&\quad - \bar{\lambda}\Psi_X^* \Psi_Y + |\lambda|^2 \Psi_X^* \Psi_X]\mathbf{v}.
\end{aligned}
\tag{3}
$$

Notice that Eq. (3) is calculated only for precomputed eigenpairs. However, the *spectral residual* serving as a threshold only filters polluted eigenpairs: it does not explore additional accurate eigenpairs or enhance the approximation. For an extreme case, the spectrum of the approximated Koopman operator could be trivial (Colbrook & Townsend, 2024, Example 3.1). This is precisely the *spectral inclusion* problem we mentioned earlier, where the computed spectrum is constrained to a subset of the true spectrum, but could potentially omit critical spectral components. Addressing this fundamental limitation is a core objective of our work.

**Continuous Spectra and Pseudospectrum** The concept of *pseudospectrum* is particularly useful when studying the continuous spectrum of the Koopman operator, as it captures regions where the resolvent norm (i.e., $\|(\mathcal{K} - \lambda I)^{-1}\|$) is large, even in the absence of discrete eigenvalues. In order to compute the pseudospectrum, a set of grid points as candidate spectrum values $\{z_1, \ldots z_{n_z}\}$ is scanned in the complex plane using the spectral residual; specifically, for each grid point $z_j \in \mathbb{C}$, we compute the following SVD problem

$$
\tau_j = \min_{\mathbf{v}_i \in \mathbb{C}^{N_k}} \widehat{res}(z_j, \mathbf{\Psi v}_i),
\tag{4}
$$

which uses the dictionary $\mathbf{\Psi}$ computed from our proposed ResKoopNet method. The approximated $\epsilon$-pseudospectrum containing the continuous spectra is then given by $\{z_j : \tau_j < \varepsilon\}$. More details on proof of convergence can be found in Colbrook & Townsend (2024, Appendix B).

### 3.2. Residual-based Koopman-learning Network (ResKoopNet)

**General framework** In this section, we present the Residual-based Koopman-learning Network (ResKoopNet), which minimizes *spectral residual* to address the spectral inclusion problem and yield a more accurate and complete Koopman spectrum. The overall pipeline is illustrated in Figure 1. ResKoopNet optimizes the dictionary functions by minimizing the total *spectral residual* $J$ as following

$$
J := \sum_{i=1}^{N_K} \widehat{res}(\lambda_i, \phi_i)^2,
$$

over all computed eigenpairs $\{(\lambda_i, \phi_i)\}_{i=1}^{N_K}$. This loss directly influences the quality of the finite-dimensional Koopman approximation and ensures the learned dictionary captures essential spectral features without relying on precomputed spectra or post-processing.

In this work, we parameterize the dictionary functions $\Psi(x; \theta)$ using a simple feedforward neural network, as shown on the right side of Figure 1, where $\theta$ denotes the network parameters. Unlike EDMD, which minimizes prediction error, our method explicitly targets spectral accuracy.

*Remark* 3.1. Notice that the neural network architecture can be replaced by more advanced models or other optimizers depending on the system and data structure.

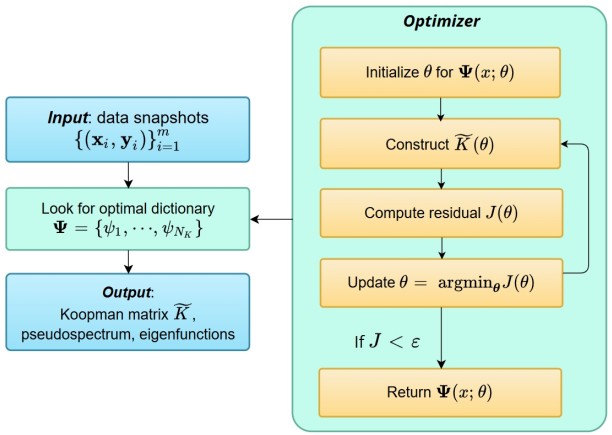

Figure 1: The optimizer on the right searches for the optimal dictionary by minimizing Eq. (3).

In ResDMD's framework, the *spectral residual* in Eq. (3) measures how well a computed eigenpair from the dataset, in other words, ideally we want the total *spectral residual* $J = \sum_{i=1}^{N_K} \widehat{res}(\lambda_i, \phi_i)^2$ approaches to zero, which implies that we can treat $J$ as a loss function. Moreover, minimizing $J$ is also equivalent to the following minimization problem:

$$\min_{\tilde{K}} J = \min_{\tilde{K}} \frac{1}{m}\|(\Psi_Y - \Psi_X \tilde{K})V\|_F^2, \qquad (5)$$

where each column of the matrix $V$ is an (right) eigenvector $\mathbf{v}_i$ of the matrix $\tilde{K}$. Thus, with dictionary $\Psi$ fixed, we can obtain a closed form for the optimal Koopman matrix $\tilde{K}$ as following

$$\tilde{K} = G^\dagger A, \qquad (6)$$

where $G = \frac{1}{m}\Psi_X^* \Psi_X$, $A = \frac{1}{m}\Psi_X^* \Psi_Y$ (See A.3 for more details on deriving the Eq. (5) and Eq. (6)).

Although Eq. (6) resembles the EDMD formulation for computing the Koopman matrix representation $\tilde{K}$, ResKoopNet and ResDMD fundamentally differ from EDMD in their theoretical foundation. While they share a similar matrix expression, this expression is derived from minimizing the *spectral residual* in Eq. (3) rather than the prediction error that EDMD minimizes, as shown in Appendix A.3. Specifically, our approach explicitly incorporates $\mathcal{K}^*\mathcal{K}$ as shown in Eq. (2), focusing on spectral accuracy rather than observable prediction. This distinction leads to significantly

different performance in capturing the spectral properties of the Koopman operator.

*Remark* 3.2. To ensure the numerical stability when computing $G^\dagger$, we add a small perturbation, i.e., $\tilde{K} = (G+\sigma I)^{-1} A$ for some small number $\sigma > 0$.

As shown in Eq. (6), ResKoopNet provides an explicit expression for the optimal Koopman matrix $\tilde{K}$ given a fixed dictionary function $\Psi$. Although the formulation is similar to EDMD, the key distinction between ResKoopNet and EDMD lies in their loss functions; more specifically, while EDMD minimizes the overall prediction error $\|\Psi_Y - \Psi_X \tilde{K}\|_F^2$, ResKoopNet minimizes the residual projected onto the estimated eigenspace, i.e., $\|(\Psi_Y - \Psi_X \tilde{K})V\|_F^2$. This eigenspace projection fundamentally changes the optimization landscape by weighting errors along spectral directions, ensuring that the learned Koopman matrix better captures the underlying eigenstructure rather than merely minimizing average prediction error. Therefore, they arise from distinct theoretical foundation.

**From Residual to Machine Learning** This section explains how a simple neural network is integrated into ResKoopNet. In the optimizer as shown in Figure 1, the neural network parameterizes the dictionary $\Psi(x; \theta)$ and minimize the parametrized total *spectral residual* $J(\theta)$ given in the following:

$$J(\theta) = \frac{1}{m}\|(\Psi_Y(\theta) - \Psi_X(\theta)K(\theta))V(\theta)\|_F^2, \qquad (7)$$

where $K(\theta)$ and $V(\theta)$ depend on the parameters $\theta$. The optimal Koopman matrix $\tilde{K}(\theta)$ is computed following Eq. (6):

$$\tilde{K}(\theta) = G(\theta)^\dagger A(\theta). \qquad (8)$$

The algorithm alternates between updating $K(\theta)$ via Eq. (8) and optimizing $J(\theta)$ w.r.t. $\theta$ by stochastic gradient descent via Eq. (7). While it is possible to optimize both $K(\theta)$ and $J(\theta)$ simultaneously, as done in Takeishi et al. (2017) and Otto & Rowley (2019), our separate procedure ensures computational efficiency and numerical stability (Li et al., 2017).

**Computing Algorithm** In our neural network implementation, we include some non-trainable functions to enhance the dictionary computation. Specifically, we add a vector of constant one and the coordinates of the state space as non-trainable basis in the output layer, which helps avoid trivial solutions, i.e., $J = 0$ for some initial $\theta$. For the network architecture, we build a simple feedforward neural network with three hidden layers, where each hidden layer size can be specified during training. We use the hyperbolic tangent (tanh) function as the activation function for the hidden layers and employ the Adam optimizer for updating the network parameters $\theta$. Adam is particularly well-suited for this task due to its ability to adapt the learning rate for

**Algorithm 1** ResKoopNet

---

**Input:** Dataset $X, Y$, number of observables $N_K$, learning step $\delta$, regularization parameter $\sigma$, loss function threshold $\epsilon > 0$, grid points $\{z_1, \ldots z_{n_z}\}$
**1:** Initialize $\theta$, thus initializing $\boldsymbol{\Psi}(\theta)$
**2:** Compute $\tilde{K}(\theta)$ and its eigenvector matrix $V(\theta)$
**repeat**
    Update $\theta = \theta - \delta \nabla_\theta J(\theta)$
    Compute $G(\theta) = \frac{1}{m} \Psi_X^* \Psi_X, A(\theta) = \frac{1}{m} \Psi_X^* \Psi_Y$
    Update $\tilde{K}(\theta) = (G(\theta) + \sigma I)^{-1} A(\theta)$ and $V(\theta)$
**until** $J(\theta) < \epsilon$
**Output:** $\tilde{K}(\theta)$, eigenpairs $\{(\lambda_i, \phi_i = \boldsymbol{\Psi} \mathbf{v}_i)\}_{i=1}^{N_K}$ and $\epsilon$-pseudospectrum $\{z_j : \tau_j < \varepsilon\}$ (See Eq. (4)).

---

each parameter, which can lead to faster convergence in the alternating optimization process between the network parameters $\theta$ and the Koopman matrix $K(\theta)$ formulation. The computing steps are illustrated in the following Algorithm 1. ResKoopNet's practical advantages come with computational demands due to its iterative optimization. Each gradient update scales linearly with system dimensionality and network parameters. While individual steps are lightweight, repeated least-squares optimizations make convergence slower than standard numerical methods, with stochastic gradient descent achieving an $O(1/n)$ rate.

We also give a brief discussion on the convergence analysis of ResKoopNet in Appendix A.4, which leverages existing results from approximation theory in the Barron space (Haykin, 2009; Weinan et al., 2019).

# 4. Application in physical and biological systems

In this section, we present three examples to demonstrate ResKoopNet's effectiveness in estimating key components of Koopman Mode Decomposition: spectrum, eigenfunctions, and Koopman modes. In the pendulum example, we compare with EDMD, EDMD-DL, Hankel-DMD, and ResDMD, and show that ResKoopNet captures the full Koopman spectrum with significantly fewer dictionary observables than required by ResDMD, while other methods either produce spectral pollution or miss the continuous spectrum. Our turbulence analysis demonstrates ResKoopNet's ability to detect not only acoustic vibration and turbulent fluctuation (as in ResDMD (Colbrook & Townsend, 2024, Section 4.3.1, Section 6.3)) but also to recover the fundamental pressure field structure through the lowest-residual Koopman mode, which is a feature not captured by Hankel-DMD using the same dataset. In the neural dynamics example, we compare against three methods across five mice datasets, with ResKoopNet achieving superior clustering of neural states corresponding to different visual stimuli, as quantified

by the Davies-Bouldin index. Note that we include Hankel-DMD (Arbabi & Mezic, 2017) as a consistent benchmark across all experiments due to its theoretical guarantees and applicability to both low and high-dimensional systems without requiring predefined dictionaries. It uses time-delayed state measurements by Takens Embedding Theorem (Takens, 2006) instead of using a pre-defined dictionary or applying Galerkin approximation as in EDMD and ResDMD methods (see Appendix A.8.1 for more details).

### 4.1. Pendulum

The pendulum system is a measure-preserving Hamiltonian system, which theoretically constrains its spectrum to the unit circle. The system is governed by:

$$\ddot{\theta} = \sin(\theta), \quad (\theta, \dot{\theta}) \in [-\pi, \pi]_{\text{per}} \times [-15, 15],$$

where $\theta$ represents the angular displacement from equilibrium and $\dot{\theta}$ the angular velocity. The dynamics exhibit two distinct regimes: oscillatory motion for low-energy initial conditions, and rotational motion when the initial energy exceeds the separatrix threshold (Lusch et al., 2018).

For our analysis, we sample 90 and 240 initial conditions uniformly distributed in the domain, each evolving for 1000 time steps with $\Delta t = 0.5$. This gives datasets of approximately $9 \times 10^4$ and $2.4 \times 10^5$ snapshots, respectively. We choose 3 hidden layers and use 300 neurons and 350 neuron in each hidden layer in both cases, respectively.

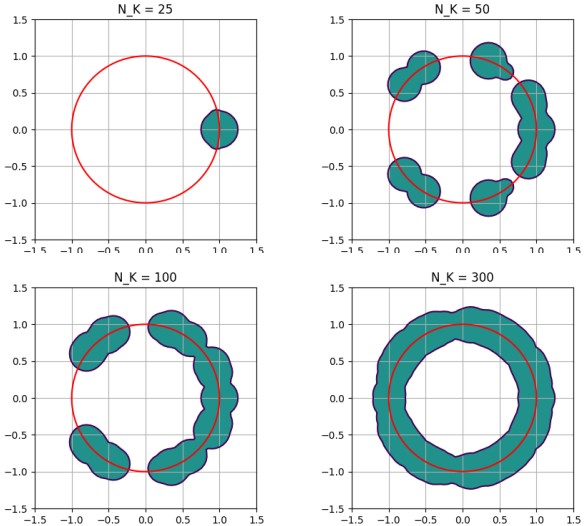

Figure 2: The four plots depict the spectrum of the Koopman operator, constructed using varying dictionary sizes $N_K$ of 25, 50, 100, and 300. Each plot utilizes 90 initial points to illustrate the impact of increasing the dictionary size on approximating the spectrum of the Koopman operator.

As shown in Figure 2, ResKoopNet requires only $N_K = 300$ observables to approximate the full spectrum. While the

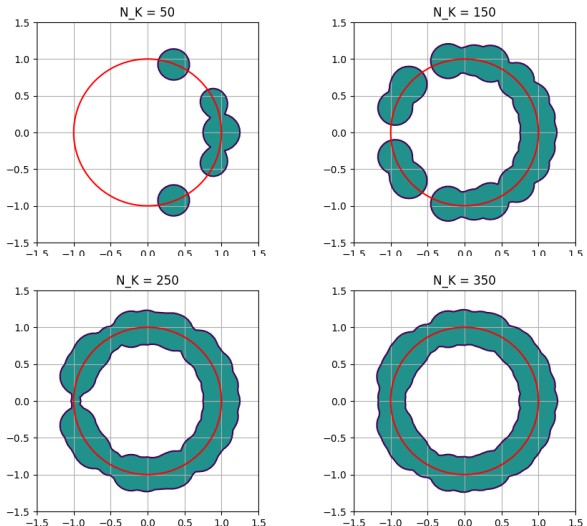

Figure 3: Same example as Figure 2 but with larger data size, using 240 initial points to show the effect of increasing dictionary size on the Koopman spectrum approximation.

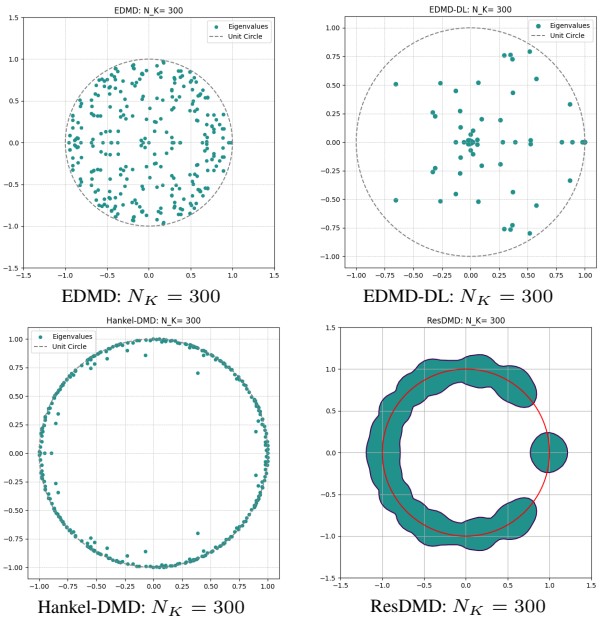

Figure 4: Comparison with classical methods. Eigenvalue spectra computed from a $300 \times 300$ Koopman matrix using EDMD, EDMD-DL, Hankel-DMD, and ResDMD.

original ResDMD work reported using 964 observables (Colbrook & Townsend, 2024, Section 4.3.1), our additional tests indicate that approximately 460 basis functions can produce reasonable circular approximations with ResDMD (see Figure 7 in Appendix A.7.1). Nevertheless, ResKoopNet still achieves comparable accuracy with significantly fewer observables. This efficiency persists even with larger sizes

of data points (Figure 3), where $N_K = 350$ observables remain sufficient. Figure 4 compares ResKoopNet with four classical methods using the 90-initial-point dataset. Both EDMD and ResDMD employ Hermite functions (order 15) for $\theta$ and Fourier functions (order 20) for $\dot{\theta}$, while Hankel-DMD uses a time delay of 150. Though Hankel-DMD produces eigenvalues near the unit circle, it only captures the discrete point spectrum and cannot reconstruct the continuous spectral components. We include Hankel-DMD here for consistency in benchmarking across all experiments, despite its limitations in higher-dimensional systems examined later. The shaded region in the ResDMD plot represents the $\epsilon$-pseudospectrum, which approximates regions where the *spectral residual* is smaller than a threshold value; however, with 300 basis functions, ResDMD fails to fully recover the unit circle. Overall, this pendulum example demonstrates that ResKoopNet achieves accurate and complete spectral recovery using much fewer observables than other classical methods.

### 4.2. Turbulence

Recovering spatial patterns is a typical goal of DMD-based methods, especially in fluid dynamics (Schmid, 2022; Mezić, 2013). Figure 5(a) shows the ground truth pressure distribution, in which a clear asymmetry exists between the upper and lower airfoil surfaces. This fundamental spatial pattern is critical for analyzing aerodynamic performance and structural integrity. While the original Kernel-ResDMD method has successfully detected acoustic vibration and turbulent fluctuation shown in Colbrook & Townsend (2024, Section 6.3), it fails to recover the global spatial structure of the pressure field. In contrast, ResKoopNet achieves a significant improvement by accurately reconstructing this dominant spatial pressure field structure itself from its leading Koopman mode as shown in Figure 5(b), which contains essential information about the flow characteristics around the airfoil.

The turbulent flow dataset from Colbrook & Townsend (2024, Section 6.3) models a two-dimensional airfoil system with Reynolds number $3.88 \times 10^5$ and Mach number 0.07. The data captures a pressure field at 295,122 spatial points across 798 time steps, sampled every $2 \times 10^{-5}$ seconds. Given the extreme dimensionality of this $X \in \mathbb{R}^{798 \times 295,122}$ matrix, we apply truncated SVD on the data matrix to retain only the top 150 components and project the data via $XV = US \in \mathbb{R}^{798 \times 150}$. After computing Koopman modes in this reduced state space using ResKoopNet, we map them back to the original dimensions through multiplication with $V$. To maintain consistency with the baseline, we use the same number of observables $N_K = 250$ as in Colbrook & Townsend (2024, Section 6.3), including 99 trainable neural-network-based functions, one constant function, and 150 principal components obtained from the truncated SVD.

Figure 5(b) shows the first Koopman mode computed by ResKoopNet, which corresponds to the mode whose eigenvalue has the largest absolute real part, and it also has the smallest *spectral residual* among all computed modes. This mode not only captures the most dominant dynamics in the system but also closely resembles the original pressure field. This correspondence demonstrates that ResKoopNet is able to recover the principal spatial feature of the turbulent flow, something that Kernel-ResDMD and Hankel-DMD could not achieve, which also justifies that the proposed method ResKoopNet can effectively address the spectral inclusion problem where traditional methods often fail to recover the complete spectrum of the Koopman operator. This mode likely corresponds to the largest singular value observed in the SVD, as shown in Figure 5(e) and (f). The dominant singular value highlights the strongest spatial mode in the data, which aligns with the physical air pressure field and its global structure.

In addition to the first mode, Figure 5(c) and (d) show the third and seventh Koopman modes computed by ResKoopNet, associated with the eigenvalues having the third and seventh largest absolute values of their real parts. These two modes have the second and third smallest spectral residuals and correspond to acoustic vibration and turbulent fluctuation, as also observed in the Kernel-ResDMD study (Colbrook & Townsend, 2024, Section 6.3). The ordering of the Koopman modes in Figure 5(b)–(d) is based on the ascending values of *spectral residual*, indicating how well each mode aligns with the spectrum of the underlying Koopman operator.

For further comparison, we include in Appendix A.8.2 four Koopman modes computed using Hankel-DMD with a time delay of 5, selected similarly based on the smallest *spectral residual*. These modes, however, do not reveal the global pressure pattern observed in ResKoopNet's first mode, which further validates the unique capability of ResKoopNet to recover large-scale coherent structures in turbulent systems.

### 4.3. Neural dynamics identification in mice visual cortex

Since ResKoopNet directly minimizes the *spectral residuals* based on eigenfunctions, its estimated evolution of eigenfunctions over time should ideally capture latent dynamics. To evaluate how effectively ResKoopNet reveals latent temporal dynamics in real data, we apply it to a dataset of high-dimensional neural signals and demonstrate its advantages over a series of classical methods: the Hankel-DMD, EDMD (combined with RBF basis) and Kernel-ResDMD. These methods are selected as representative approaches for handling high-dimensional data.

The dataset is part of the open dataset on mice from the competition "Sensorium 2023" (Turishcheva et al., 2024b;a). In

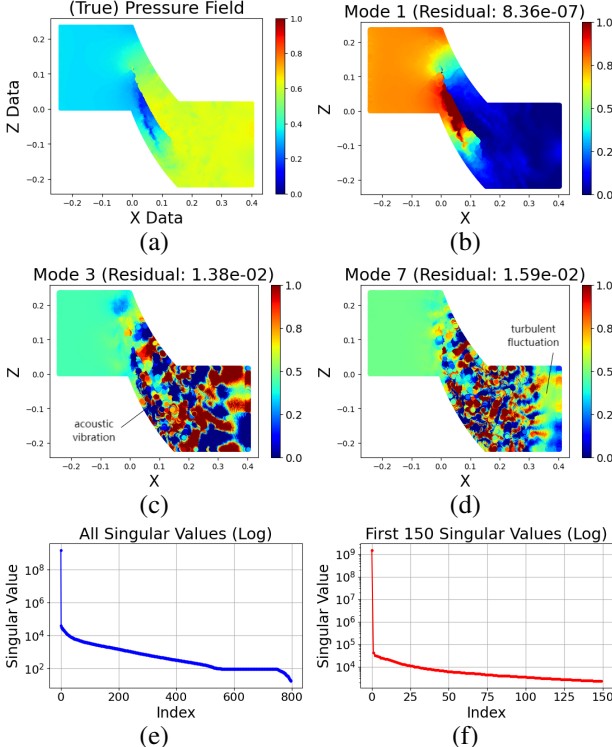

Figure 5: Turbulence detection using Koopman modes from 250 observables. (a) shows the original 2D pressure field. (b) shows Koopman mode 1 which has the smallest residual and closely matches the original pressure field. (c) and (d) show the acoustic vibration and turbulent fluctuation by the Koopman modes with the 2nd and 3rd smallest spectral residual. (e) and (f) show the plot of all singular values and first 150 singular values when applying truncated SVD method, respectively.

the experiments, mice viewed natural videos while their neural signals were recorded via calcium imaging in the primary visual cortex, reflecting the activity of thousands of neurons. Here, we focus on the state partitioning of neural signals. Specifically, in each mouse, six video stimuli were repeatedly shown, creating ideal conditions to define brain states. Neural activity during repeated trials with the same stimuli is assumed to reflect the same underlying dynamic system, enabling Koopman decomposition methods to uncover and separate these brain states.

The dataset consists of neural recordings from five mice, each exposed to 6 video stimuli, repeated 9-10 times for a total of around 60 trials. Each recording captures the activity of over 7,000 neurons, with each 10-second video sampled at 50 Hz, resulting in 300 data points per trial.

We applied ResKoopNet and three classical Koopman decomposition methods (Hankel-DMD, EDMD with RBF basis, and Kernel-ResDMD) to this dataset, using different

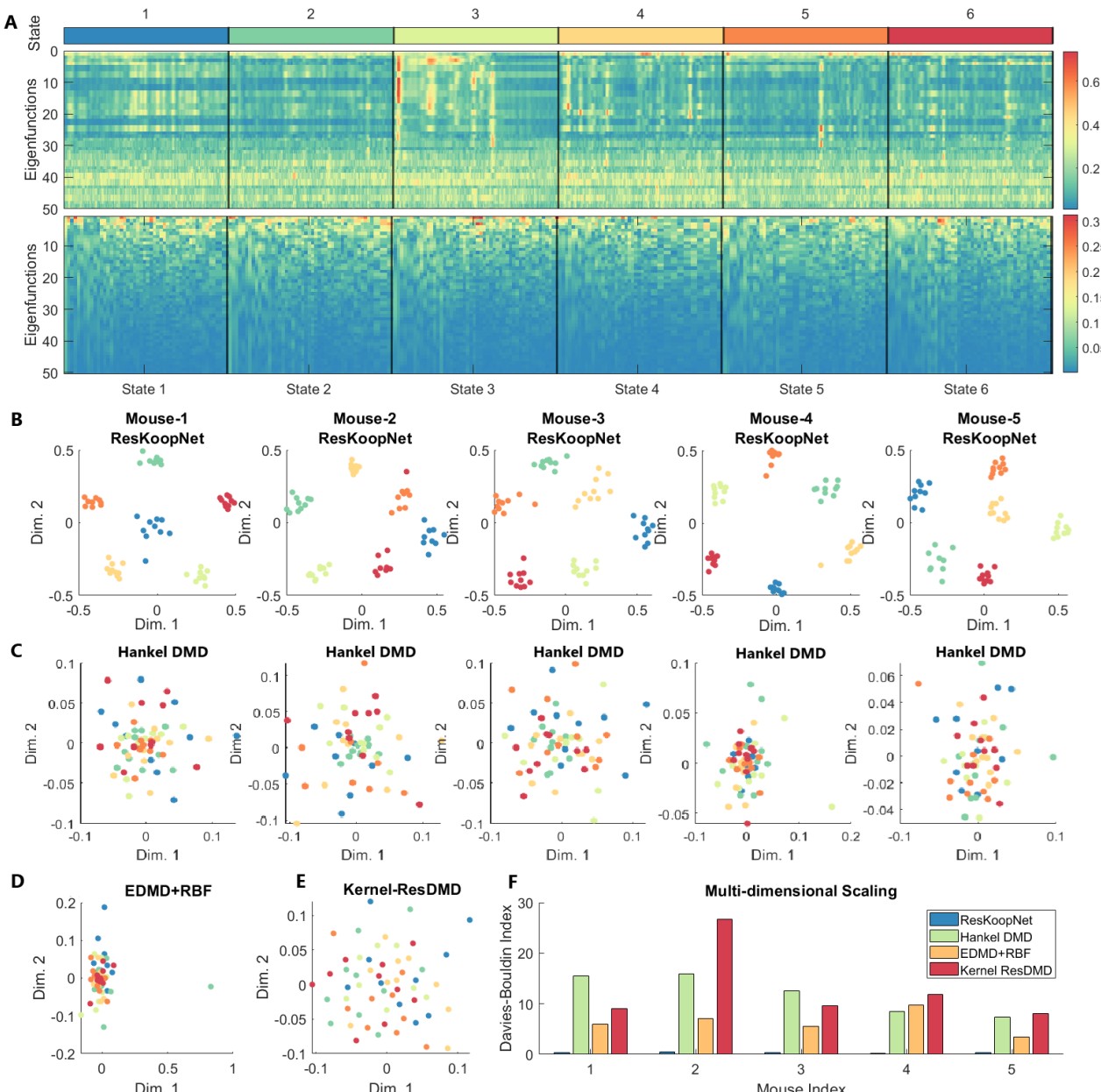

Figure 6: ResKoopNet outperforms Hankel-DMD in identifying latent dynamic structures in neural signals with a dictionary size of 50. (A) (Top) 50 Koopman eigenfunctions estimated by ResKoopNet across 6 states characterized by different video stimuli in an example mouse. Each trial contains 300 data points (10s at 50Hz). (Bottom) 50 Koopman eigenfunctions approximated by Hankel-DMD, each 50 points long, reflecting the dimension of the Hankel matrix. (B) 2D representation of Koopman eigenfunctions for all tested mice, computed by ResKoopNet and reduced via Multidimensional Scaling (MDS). Trials of the same state cluster well. (C) Same as (B) but computed with Hankel-DMD, showing no clear state separation. (D) 2D representation of Koopman eigenfunctions for the first mouse, computed by EDMD with an RBF basis. See Appendix Figure 14 for full results. (E) Same as (D) but computed with Kernel-ResDMD. See Appendix Figure 15 for full results. (F) Davies-Bouldin Indices (DBIs) evaluating clustering quality across four methods (ResKoopNet, Hankel-DMD, EDMD+RBF, and Kernel-ResDMD) for five mice. Lower DBI values for ResKoopNet indicate better clustering.

implementations and Koopman subspace dimensions. For ResKoopNet, we trained dictionaries on all snapshots from each mouse to avoid overfitting. We reduced the data to 24 dimension via truncated SVD, together with the constant and 25 trainable bases, resulting in 50 eigenfunctions. The neural network we use here consists of 3 hidden layers, each with 200 neurons. The hyperparameter scanning results are included in Appendix Figure 17, which demonstrates that our parameter choice is within the robust range. The decomposed eigenfunctions are shown in Figure 6A(top), with markers indicating ground truth state separations. For Hankel-DMD, we built a Hankel matrix with a delay of 50, producing 50 eigenfunctions per trial. In EDMD with RBF basis, we used the SVD-truncated 300 basis and 1000 RBF functions, resulting in 1301 eigenfunctions. For Kernel-ResDMD, we used normalized Gaussians as kernel functions, setting the Koopman subspace dimension to 299 eigenfunctions based on Colbrook et al. (2023). See Appendix A.9.4 for method details and Appendix A.10 for dictionary size justification. These eigenfunctions, shown in Figure 6A(bottom), Appendix Figure 14A, and Appendix Figure 15A, are compared to ground truth trial identities.

In this study, Koopman eigenfunctions are interpreted as representing dynamical features corresponding to the video stimuli. Consequently, their effectiveness in capturing these key, stimulus-related dynamics is evaluated by assessing how well eigenfunctions from trials with the same video stimulus cluster together and remain distinguishable from those associated with different video stimuli. This effectively transforms the problem into a clustering task based on the separability of eigenfunctions across different stimuli. It is noteworthy that for the ResKoopNet case, averaged trial differences are visibly clear. It is also worth clarifying that clustering these processes is not biologically necessary, as simple clustering of mean firing rates is sufficient (see Appendix Figures 10, 11, 12). However, this dataset provides an ideal benchmark with clear ground truth labels, enabling accurate comparison of Koopman eigenfunction estimation methods. Effective estimation of Koopman eigenfunctions can be valuable for more complex tasks, such as unsupervised latent state identification or decoding object dynamics from video stimuli, which offers promising directions for future research.

We use Multi-dimensional Scaling (MDS) to visualize how these eigenfunction-based features cluster according to ground truth states. MDS reduces data dimensionality based on similarities, making it ideal for visualizing clustering performance. While Uniform Manifold Approximation and Projection (UMAP) and t-distributed Stochastic Neighbor Embedding (t-SNE) are alternative methods, we show MDS results in 2D space (Figure 6B-E), with similar results for UMAP and t-SNE in Appendix Figure 13, Appendix Figure 14C, D and Appendix Figure 15C, D.

The 2D MDS visualization reveals a clear separation of features for all 5 mice using ResKoopNet (Figure 6B), whereas no other method shows clear clustering (Figure 6C-E, Appendix Figure 14B, Appendix Figure 15B). To quantify this clustering, we calculate the Davies-Bouldin index (DBI), a measure of clustering quality that assesses how compact and well-separated the clusters are. A lower DBI indicates more compact clusters that are farther apart from each other, which corresponds to better clustering. The DBI is significantly lower for ResKoopNet (Figure 6F), suggesting that it captures the latent dynamic structure more effectively than all three other methods. Similar clustering patterns are confirmed with UMAP and t-SNE (Appendix Figure 16).

## 5. Conclusion and future work

In this paper, we introduced ResKoopNet, a novel approach that minimizes *spectral residuals* to accurately approximate Koopman operators for complex dynamical systems. This method successfully resolves the spectral inclusion problem while capturing both discrete and continuous spectra with high accuracy. Our experiments across physical and biological systems demonstrate that ResKoopNet significantly outperforms existing methods in identifying coherent structures and uncovering latent dynamics, particularly in high-dimensional applications.

Despite these advantages, ResKoopNet has several limitations. First, the use of neural networks increases computational cost compared to classical numerical algorithms (see Appendix A.6). Second, unlike stochastic approaches such as VAMP (Mardt et al., 2018) and SDMD (Xu et al., 2025), ResKoopNet currently does not account for stochasticity (see Appendix A.5 for comparison discussion). In addition, its performance can vary depending on the neural architecture and training configuration.

There are several promising directions to address these limitations and extend ResKoopNet's capabilities. Many physical laws, such as conservation principles or symmetry constraints, are inherently encoded in the spectral properties of dynamical systems. Incorporating such domain knowledge directly into the neural network architecture could significantly enhance the learning of Koopman spectral information. Future work could explore physics-informed neural network designs, develop efficient training strategies to reduce computational costs, and extend the framework to handle stochastic dynamics. Ultimately, ResKoopNet and its future variants have the potential to become powerful tools for understanding and predicting the behavior of complex dynamical systems across diverse scientific and engineering applications, and bridge the gap between data-driven methods and physical principles.

## Acknowledgements

Authors are grateful to Matthew Colbrook for valuable consultation on theoretical aspects of ResDMD and providing fluid data, to Bin Han for discussion on computational issues, to Rouslan Krechetnikov for discussion on turbulence analysis, to Michel Beserve and Rory Bufacchi for proofreading and to Turishcheva Polina for providing permission of using the Sensorium 2023 dataset.

## Impact Statement

This work contributes to advancing Koopman operator learning through neural network-based *spectral residual* minimization. ResKoopNet enhances data-driven analysis in fields such as physics, engineering, and neuroscience by capturing continuous spectral structures and enabling automatic dictionary exploration. Future work could explore implementation with physical constraints and extensions to stochastic systems. No direct ethical concerns are anticipated, as this research primarily focuses on data-driven methods for dynamical systems analysis.

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

# A. Appendix

## A.1. Source Code

For reproducibility, the source code is available at this link.

## A.2. Calculation steps for Eq. (3)

Here we are going to show how *spectral residual* implies (3).

Consider $\phi = \mathbf{\Psi v} = \sum_{i=1}^{N_K} \psi_i \mathbf{v}_i$ with $\|\phi\|_\mu = 1$, then

$$\frac{\int_\Omega |\mathcal{K}\phi(x) - \lambda\phi(x)|^2 d\mu(x)}{\int_\Omega |\phi(x)|^2 d\mu(x)}$$

$$= \int_\Omega |\mathcal{K}\phi(x) - \lambda\phi(x)|^2 d\mu(x)$$

$$= \langle \mathcal{K}\phi - \lambda\phi, \mathcal{K}\phi - \lambda\phi \rangle_\mu$$

$$= \langle \mathcal{K}\phi, \mathcal{K}\phi \rangle_\mu - \langle \lambda\phi, \mathcal{K}\phi \rangle_\mu - \langle \mathcal{K}\phi, \lambda\phi \rangle_\mu + \langle \lambda\phi, \lambda\phi \rangle_\mu$$

$$= \langle \mathcal{K}\mathbf{\Psi v}, \mathcal{K}\mathbf{\Psi v} \rangle_\mu - \bar{\lambda}\langle \mathbf{\Psi v}, \mathcal{K}\mathbf{\Psi v} \rangle_\mu - \lambda\langle \mathcal{K}\mathbf{\Psi v}, \mathbf{\Psi v} \rangle_\mu + |\lambda|^2 \langle \mathbf{\Psi v}, \mathbf{\Psi v} \rangle_\mu$$

$$= \langle \sum_{i=1}^{N_K} \mathcal{K}\psi_i \mathbf{v}_i, \sum_{j=1}^{N_K} \mathcal{K}\psi_j \mathbf{v}_j \rangle_\mu - \bar{\lambda}\langle \sum_{i=1}^{N_K} \psi_i \mathbf{v}_i, \sum_{j=1}^{N_K} \mathcal{K}\psi_j \mathbf{v}_j \rangle_\mu - \lambda\langle \sum_{i=1}^{N_K} \mathcal{K}\psi_i \mathbf{v}_i, \sum_{j=1}^{N_K} \psi_j \mathbf{v}_j \rangle_\mu + |\lambda|^2 \langle \sum_{i=1}^{N_K} \psi_i \mathbf{v}_i, \sum_{j=1}^{N_K} \psi_j \mathbf{v}_j \rangle_\mu$$

$$= \sum_{i,j=1}^{N_K} \bar{\mathbf{v}}_i \langle \mathcal{K}\psi_i, \mathcal{K}\psi_j \rangle_\mu \mathbf{v}_j - \bar{\lambda} \sum_{i,j=1}^{N_K} \bar{\mathbf{v}}_i \langle \psi_i, \mathcal{K}\psi_j \rangle_\mu \mathbf{v}_j - \lambda \sum_{i,j=1}^{N_K} \bar{\mathbf{v}}_i \langle \mathcal{K}\psi_i, \psi_j \rangle_\mu \mathbf{v}_j + |\lambda|^2 \sum_{i,j=1}^{N_K} \bar{\mathbf{v}}_i \langle \psi_i, \psi_j \rangle_\mu \mathbf{v}_j$$

$$= \sum_{i,j=1}^{N_K} \bar{\mathbf{v}}_i \left[ \langle \mathcal{K}\psi_i, \mathcal{K}\psi_j \rangle_\mu - \bar{\lambda}\langle \psi_i, \mathcal{K}\psi_j \rangle_\mu - \lambda\langle \mathcal{K}\psi_i, \psi_j \rangle_\mu + |\lambda|^2 \langle \psi_i, \psi_j \rangle_\mu \right] \mathbf{v}_j$$

$$\approx \sum_{i,j=1}^{N_K} \bar{\mathbf{v}}_i \left[ \frac{1}{m}[\Psi_Y^* \Psi_Y]_{ij} - \bar{\lambda}\frac{1}{m}[\Psi_X^* \Psi_Y]_{ij} - \lambda\frac{1}{m}[\Psi_Y^* \Psi_X]_{ij} + |\lambda|^2 \frac{1}{m}[\Psi_X^* \Psi_X]_{ij} \right] \mathbf{v}_j$$

$$= \frac{1}{m}\mathbf{v}^* \left[ \Psi_Y^* \Psi_Y - \lambda(\Psi_X^* \Psi_Y)^* - \bar{\lambda}\Psi_X^* \Psi_Y + |\lambda|^2 \Psi_X^* \Psi_X \right] \mathbf{v}$$

$$\left( = \text{Eq. (3)} \right)$$

### A.3. Details for deriving (5)

$$J = \sum_{i=1}^{N_K} \widehat{res}(\lambda_i, \phi_i)^2$$

$$= \sum_{i=1}^{N_K} \frac{1}{m} \mathbf{v}_i^* \left[ \Psi_Y^* \Psi_Y - \lambda_i (\Psi_X^* \Psi_Y)^* - \bar{\lambda}_i \Psi_X^* \Psi_Y + |\lambda_i|^2 \Psi_X^* \Psi_X \right] \mathbf{v}_i$$

$$= \sum_{i=1}^{N_K} \frac{1}{m} \left[ \mathbf{v}_i^* (\Psi_Y^* \Psi_Y) \mathbf{v}_i - \mathbf{v}_i^* (\Psi_X^* \Psi_Y)^* \lambda_i \mathbf{v}_i - \mathbf{v}_i^* \bar{\lambda}_i (\Psi_X^* \Psi_Y) \mathbf{v}_i + \mathbf{v}_i^* K^* (\Psi_X^* \Psi_X) K \mathbf{v}_i \right]$$

$$= \sum_{i=1}^{N_K} \frac{1}{m} \left[ \mathbf{v}_i^* (\Psi_Y^* \Psi_Y) \mathbf{v}_i - \mathbf{v}_i^* (\Psi_X^* \Psi_Y)^* K \mathbf{v}_i - \mathbf{v}_i^* K^* (\Psi_X^* \Psi_Y) \mathbf{v}_i + \mathbf{v}_i^* K^* (\Psi_X^* \Psi_X) K \mathbf{v}_i \right]$$

$$= \sum_{i=1}^{N_K} \frac{1}{m} \left( \langle \Psi_Y \mathbf{v}_i, \Psi_Y \mathbf{v}_i \rangle - \langle \Psi_Y \mathbf{v}_i, \Psi_X K \mathbf{v}_i \rangle \right.$$

$$\left. - \langle \Psi_X K \mathbf{v}_i, \Psi_Y \mathbf{v}_i \rangle + \langle \Psi_X K \mathbf{v}_i, \Psi_X K \mathbf{v}_i \rangle \right)$$

$$= \sum_{i=1}^{N_K} \frac{1}{m} \langle \Psi_Y \mathbf{v}_i - \Psi_X K \mathbf{v}_i, \Psi_Y \mathbf{v}_i - \Psi_X K \mathbf{v}_i \rangle$$

$$= \sum_{i=1}^{N_K} \frac{1}{m} \| \Psi_Y \mathbf{v}_i - \Psi_X K \mathbf{v}_i \|_\mu^2$$

$$= \frac{1}{m} \| (\Psi_Y - \Psi_X K) V \|_F^2.$$

Next, by matrix calculus with denominator layout convention, we try to find minimal of $J$:

$$0 = \frac{dJ}{dK} = \frac{d \operatorname{tr}(J)}{dK} \quad \text{(since } J \text{ is a scalar)}$$

$$= \frac{d}{dK} \operatorname{tr} \left( \frac{1}{m} \sum_{i=1}^{N_K} \mathbf{v}_i^* \left[ \Psi_Y^* \Psi_Y - (\Psi_X^* \Psi_Y)^* K - K^* (\Psi_X^* \Psi_Y) + K^* (\Psi_X^* \Psi_X) K \right] \mathbf{v}_i \right)$$

$$= \sum_{i=1}^{N_K} \frac{d}{dK} \operatorname{tr} \left( \mathbf{v}_i^* \left[ L - A^* K - K^* A + K^* G K \right] \mathbf{v}_i \right)$$

$$= \sum_{i=1}^{N_K} \frac{d}{dK} \operatorname{tr} (\mathbf{v}_i^* L \mathbf{v}_i) + \frac{d}{dK} \operatorname{tr} (\mathbf{v}_i^* A^* K \mathbf{v}_i) + \frac{d}{dK} \operatorname{tr} (\mathbf{v}_i^* K^* A \mathbf{v}_i) + \frac{d}{dK} \operatorname{tr} (\mathbf{v}_i^* K^* G K \mathbf{v}_i)$$

$$= \sum_{i=1}^{N_K} -A \mathbf{v}_i \mathbf{v}_i^* - A \mathbf{v}_i \mathbf{v}_i^* + (G + G^*) K \mathbf{v}_i \mathbf{v}_i^*$$

$$= \sum_{i=1}^{N_K} (-2A + 2GK) \mathbf{v}_i \mathbf{v}_i^* \quad \text{(} G \text{ is symmetric)}$$

where $\operatorname{tr}()$ is trace of a matrix and $G = \Psi_X^* \Psi_X, A = \Psi_X^* \Psi_Y, L = \Psi_Y^* \Psi_Y$.

Since eigenvector $v_i$ is not a zero vector, $v_i v_i^*$ is not a zero matrix. So

$$-2A + 2GK = 0 \Rightarrow K = G^\dagger A.$$

*Remark* A.1. To solve $\frac{d}{dK} \operatorname{tr} (\mathbf{v}_i^* K^* G K \mathbf{v}_i)$, we simply rewrite it as

$$\frac{d}{dK} \operatorname{tr} (\mathbf{v}_i^* K^* G K \mathbf{v}_i) = \frac{d}{dK} \operatorname{tr} ((K \mathbf{v}_i)^* G (K \mathbf{v}_i)).$$

## A.4. Discussion on convergence analysis

To understand how neural networks enhance ResKoopNet and build up the theoretical framework of convergence, it is important to first introduce Barron space (Pinkus, 1999; Cybenko, 1989; Haykin, 2009; Barron, 1993). Barron space characterizes functions efficiently approximated by two-layer neural networks, which is central to ResKoopNet. By leveraging networks that approximate functions within this space, ResKoopNet can flexibly optimize the dictionary functions for Koopman operator approximation, making it highly effective for complex, high-dimensional systems.

A function $f$ belongs to Barron space $\mathcal{B}$ if it can be represented as:

$$f(x) = \int_{\Omega} a\sigma(w^T x)\rho(da, dw),$$

where $\sigma$ is the activation function, $w$ is a weight vector, $a$ is a coefficient, and $\rho$ is a probability distribution. The complexity of $f$ is measured by the Barron norm $\|f\|_{\mathcal{B}}$:

$$\|f\|_{\mathcal{B}} = \inf_{\rho \in P_f} \left( \int_{\Omega} |a| \|w\|_1 \rho(da, dw) \right),$$

where $P_f$ is the set of distributions for which $f$ can be represented. This framework provides a foundation for analyzing approximation errors in neural networks.

The following theorem (Weinan et al., 2020) discusses the approximation capabilities of two-layer neural networks within this context, establishing a foundation for the subsequent analysis.

**Theorem A.2** (Direct Approximation Theorem, $L^2$-version). *For any $f \in \mathcal{B}$ and $r \in \mathbb{N}$, there exists a two-layer neural network $f_r$ with $r$ neurons $\{(a_i, \mathbf{w}_i)\}$ such that*

$$\|f - f_r\|_{\mu} \lesssim \frac{\|f\|_{\mathcal{B}}}{\sqrt{r}}.$$

This implies an approximation error decreasing at a rate of $O(1/\sqrt{r})$, where $r$ is the number of neurons. In ResKoopNet, the dictionary $\Psi(x; \theta) = \{\psi_i(x; \theta)\}_{i=1}^{N_K}$ is parameterized by a neural network with parameters $\theta$. Assuming the true dictionary functions $\psi_i \in \mathcal{B}$, Theorem A.2 ensures that $\Psi(x; \theta)$ can approximate the optimal dictionary spanning the Koopman invariant subspace $\mathcal{B}_{N_K} \subset \mathcal{F}$ with error $\epsilon > 0$, provided $r$ is sufficiently large.

*Remark* A.3. Notice that, the "two-layer neural network" in the Theorem A.2 statement refers to a hidder layer + an output layer, which is the most standard and general setting. Our implementation uses three hidden layers. This does not invalidate the result, as deeper networks can achieve at least the same approximation power (Pinkus, 1999).

We want to show two convergence results here: (1) $\Psi(x; \theta)$ approaches the true invariant subspace of $\mathcal{K}$; (2) The eigenpairs $(\lambda_i, \phi_i)$ and pseudospectrum approximate $\mathcal{K}$'s true spectrum as $J(\theta) \to 0$.

**Assumption A.4.** To formalize convergence, we make the following assumptions:

(a) The optimal dictionary functions $\{\psi_i^*\}_{i=1}^{N_K}$ spanning $\mathcal{K}$'s invariant subspace lie in $\mathcal{B}$.

(b) The loss $J(\theta)$ is strongly convex and Lipschitz continuous in $\theta$, and the neural network $\Psi(x; \theta)$ has bounded gradients, facilitating gradient-based optimization.

Now, consider a Barron space $\mathcal{B}$ which is dense in $\mathcal{F}$. Given $N_K$ fixed, let $\mathcal{B}_{N_K} = \text{span}\{\psi_i^*\}_{i=1}^{N_K}$ be the true invariant subspace. By Theorem A.2, for each $\psi_i^*$, there exists a neural network approximation $\psi_i(x; \theta_r)$ with $r$ neurons such that:

$$\|\psi_i^* - \psi_i(\cdot; \theta_r)\|_{\mu} \leq \frac{\|\psi_i^*\|_{\mathcal{B}}}{\sqrt{r}}.$$

The total dictionary error is:

$$\|\Psi^* - \Psi(\cdot; \theta_r)\|_F^2 = \sum_{i=1}^{N_K} \|\psi_i^* - \psi_i(\cdot; \theta_r)\|_{\mu}^2 \leq \frac{1}{r} \sum_{i=1}^{N_K} \|\psi_i^*\|_{\mathcal{B}}^2 = \frac{C_{N_K}}{r},$$

where $C_{N_K} = \sum_{i=1}^{N_K} \|\psi_i^*\|_{\mathcal{B}}^2$ is finite under Assumption A.4(a). As $r \to \infty$, $\Psi(x; \theta_r) \to \Psi^*$ in the Frobenius norm, which ensures the dictionary approximated by neural network can represent $\mathcal{B}_{N_K}$.

Algorithm 1 updates $\theta$ via stochastic gradient descent (SGD) with step size $\delta$ and computes $\tilde{K}(\theta_n)$ iteratively until $J(\theta_n) < \epsilon$ where $\theta_n$ represents $n$-th iteration of parameters $\theta$. For a Lipschitz continuous $J(\theta)$ with constant $L$ (by Assumption A.4(b)) and a strongly convex region near the optimum $\theta^*$ (assumed locally for simplicity), SGD converges at a rate of $O(1/n)$ in expectation (Bottou et al., 2018):

$$\mathbb{E}[J(\theta_n) - J(\theta^*)] \leq \frac{L}{2\eta n},$$

where $\eta$ is the strong convexity constant and $n$ is the iteration number. In practice, $J(\theta)$ is non-convex due to the neural network, and the alternating update with $\tilde{K}(\theta)$ stabilizes the process. As iteration step $n \to \infty$ and amount of data points $m \to \infty$, $J(\theta_n) \to 0$, which implies $\widehat{\mathrm{res}}(\lambda_i, \phi_i) \to 0$ for all $i$.

With $J(\theta) \to 0$, we now assess the spectral error: if $\widehat{\mathrm{res}}(\lambda_i, \phi_i) < \epsilon$, then $\|\mathcal{K}\phi_i - \lambda_i\phi_i\|_\mu / \|\phi_i\|_\mu < \sqrt{\epsilon}$, indicating $\lambda_i$ and $\phi_i$ are approximate eigenpairs of $\mathcal{K}$ converges to $\mathcal{K}$'s true spectrum as $N_K \to \infty$ and $\epsilon \to 0$, as pointed out in (Colbrook & Townsend, 2024, Theorem B.1). Uniform convergence on compact subsets of $\mathbb{C}$ follows from the density of $\mathcal{B}_{N_K}$ in $\mathcal{F}$ and Dini's theorem, as pointed out in Colbrook & Townsend (2024, Lemma B.1).

The non-convexity and system complexity may slow the spectral approximation in practice. High-order convergence (e.g., polynomial) could arise for smooth dynamics (See Colbrook & Townsend (2024, Theorem 3.1) for more details), which is useful for further study.

*Remark* A.5. The computational cost (Appendix A.6) scales with $r$, $t$, and $m$, which trades off with accuracy. Adaptive selection of $r$ and early stopping when $J(\theta) < \epsilon$ can optimize this balance.

## A.5. Highlights of ResKoopNet compared with typical existing neural network-based Koopman framework

Our ResKoopNet method takes a fundamentally different approach from existing deep learning methods by building upon the residual-based framework of ResDMD rather than the different Koopman-approximating loss functions following the variational principles of VAMPnets (Tian & Wu, 2021; Wu & Noé, 2020; Mardt et al., 2018) or the deep autoencoder structure in (Lusch et al., 2018). By incorporating *spectral residual* measures into deep learning and introducing a structured representation that captures dependencies among eigenvalues, we achieve more compact and interpretable models for nonlinear systems with continuous spectra. This approach enables us to directly minimize Koopman spectral approximation errors while avoiding the high-dimensional representations or point-spectrum limitations of previous methods.

If we take the VAMP framework as an example, here are the connections and differences. The proposed loss function and the VAMP score share the goal of optimizing approximations of the Koopman operator's spectral properties, establishing a connection in their ultimate purpose. However, although they both depend on the covariance matrices (in our manuscript Equation 3.2), their methodologies differ significantly. Our residual-based method directly minimizes the spectral approximation error of the Koopman operator and accommodates both point and continuous spectra, while the VAMP score follows a variational framework, maximizing the sum of singular values to approximate the point spectrum, primarily for stochastic systems (though see an exception in Tian & Wu (2021)). Moreover, while VAMP is specifically designed for Markov processes and requires the Koopman operator to be Hilbert-Schmidt, our approach focuses on deterministic systems and enables a more comprehensive spectral analysis that incorporates continuous spectra. This distinction in scope and methodology highlights how the two frameworks complement each other in addressing different aspects of spectral estimation.

## A.6. Discussion of computation costs

Despite the various advantages of the ResKoopNet framework, one significant limitation is its higher computational cost compared to the original ResDMD and other classical methods.

**Theoretical Perspective:** The ResKoopNet algorithm's computational demands stem primarily from its iterative optimization process. Each iteration involves a gradient descent update with complexity scaling linearly with both system dimensionality and neural network parameters. Though individual gradient steps are computationally lightweight for standard network architectures, the algorithm's efficiency issue lies in its repeated least-squares optimizations. Below we provide a detailed comparison between computation costs of several methods we used in this paper.

**Comparison over methods:** The computational costs of EDMD, ResDMD, EDMD-DL, Hankel-DMD, and ResKoopNet vary significantly based on their core computational steps and specific configurations. For a dataset with $m = 10^5$ data points and $N_K = 300$ dictionary functions (for EDMD-based methods), the theoretical complexity and runtime differ across methods. **EDMD** involves least squares and eigenvalue decomposition, with a complexity of $O(N_K^2 m + N_K^3)$, making it the fastest method, and typically requiring only seconds to minutes for computation. **ResDMD** extends EDMD by adding residual evaluation and pseudospectrum computation. The residual evaluation introduces an additional $O(N_K^3)$, and pseudospectrum computation across $n_z$ grid points incurs $O(n_z N_K^3)$, resulting in a total complexity of $O(N_K^2 m + n_z N_K^3)$. This leads to runtimes ranging from minutes to hours, depending on the resolution of the pseudospectrum grid. **EDMD-DL** incorporates dictionary learning through stochastic gradient descent (SGD), where each iteration involves matrix construction ($O(N_K^2 m)$), Koopman matrix computation ($O(N_K^3)$), and neural network forward/backward propagation ($O(|\theta|)$, with $|\theta|$ representing the total network parameter size). With $k$ SGD iterations, the total complexity becomes $O(k(N_K^2 m + N_K^3 + |\theta|))$, leading to runtimes also in the range of minutes to hours depending on $k$. **ResKoopNet**, which builds on EDMD-DL, shares the same complexity, $O(k(N_K^2 m + N_K^3 + |\theta|))$, but includes the explicit use of Koopman matrix eigenvectors and optional pseudospectrum computation, making its runtime slightly longer than EDMD-DL for high-resolution spectral analysis. **Hankel-DMD**, using a time delay embedding dimension $T$, constructs a Hankel matrix ($O(Tm)$), performs singular value decomposition (SVD) ($O(T^2 m)$), and computes the eigenvalues of a reduced $T \times T$ matrix ($O(T^3)$). The total complexity is $O(Tm + T^2 m + T^3)$, and the runtime is heavily influenced by $T$, typically ranging from minutes to hours. While EDMD is computationally the most efficient, ResDMD and Hankel-DMD provide higher precision and robustness in spectral analysis at the expense of increased runtime, and EDMD-DL and ResKoopNet offer flexibility and accuracy through dictionary learning, with additional SGD iterations and optional pseudospectrum computation contributing to their computational burden.

**Empirical Perspective:** In our experiments, without computing the pseudospectrum, the computational cost of ResDMD typically ranges from seconds to minutes. ResKoopNet, on the other hand, can require tens of minutes to several hours, depending on factors such as data dimensionality, the number of snapshots, hidden layer configurations, dictionary sizes, and training convergence criteria.

**Trade-off Between Cost and Accuracy:** While ResKoopNet's additional computational steps introduce higher costs, they enhance the accuracy and robustness of Koopman eigenpair estimation by allowing automatic dictionary learning and minimizing spurious spectral components. This trade-off makes ResKoopNet particularly valuable in applications where precision is critical. However, its computational demands render it less suitable for real-time or online Koopman model learning tasks.

**Detailed Analysis of Computational Bottlenecks:** The complexity $O(k(N_K^2 m + N_K^3 + |\theta|))$ arises from ResKoopNet's iterative optimization, alternating between updating the neural network parameters $\theta$ and computing the Koopman matrix $\hat{K}(\theta)$. The dominant term $O(kN_K^2 m)$ reflects the cost of constructing data matrices $\Psi_X(\theta)$ and $\Psi_Y(\theta)$ at each iteration, followed by gradient computation. For high-dimensional systems (large $m$) or complex architectures (large $|\theta|$), the number of iterations $k$ required for convergence increases, particularly if the loss $J(\theta) = \frac{1}{m}\|(\Psi_Y(\theta) - \Psi_X(\theta)K(\theta))V(\theta)\|_F^2$ is non-convex or ill-conditioned. The term $O(kN_K^3)$ stems from solving $\hat{K}(\theta) = (G(\theta) + \sigma I)^{-1}A(\theta)$, where matrix inversion scales cubically with $N_K$. The optional pseudospectrum computation, with complexity $O(n_z N_K^3)$, becomes significant for systems with continuous spectra, as $n_z$ grid points are scanned to evaluate residuals $\tau_j = \min_{\mathbf{v}_i} \overline{\text{res}}(z_j, \Psi \mathbf{v}_i)$. Comparatively, ResDMD's lower cost ($O(N_K^2 m + n_z N_K^3)$) avoids iteration, but sacrifices dictionary adaptability.

To mitigate these costs, we will consider some methods: (1) We consider *Mini-Batch SGD* since it uses a batch size $b \ll m$ that reduces per-iteration complexity to $O(N_K^2 b + N_K^3 + |\theta|)$, potentially lowering overall cost despite increased $k$, as validated by Barron space convergence properties. (2) We can use *preconditioning* to improve the condition number of $G(\theta)$ via diagonal scaling or incomplete Cholesky factorization could accelerate convergence, reducing $k$.

**Theoretical Extensions:** The $O(1/n)$ SGD rate assumes convexity, yet $J(\theta)$'s non-convexity suggests sublinear convergence. A bound like $\|\mathcal{K} - \mathcal{K}_{N_K}\|$ (Appendix A.4) could quantify accuracy, while determining minimal $N_K$ and $k$ for spectral completeness remains a future work. Extending ResKoopNet to stochastic systems (e.g., adding noise terms to spectral residual) or embedding physical constraints (e.g., unitary spectra for Hamiltonian systems) could enhance its scope, respectively, which also aligns with future work outlined in the Section 5.

## A.7. Extra tests for ResDMD and ResKoopNet

### A.7.1. RESDMD WITH LESS DICTIONARY FUNCTIONS

While the original ResDMD implementation used 964 basis functions to capture the full spectrum of the nonlinear pendulum, our additional tests demonstrate that the unit circle spectrum can be reasonably approximated by ResDMD with fewer observables. Figure 7 shows the progressive improvement in spectral approximation as the dictionary size $N_K$ increases. With nearly 500 basis functions, ResDMD begins to adequately reconstruct the unit circle structure, though still requiring substantially more observables than the 300 basis functions needed by ResKoopNet.

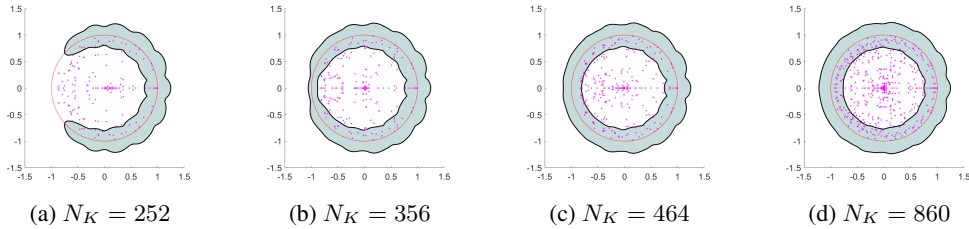

(a) $N_K = 252$     (b) $N_K = 356$     (c) $N_K = 464$     (d) $N_K = 860$

Figure 7: Comparison across different dictionary sizes by ResDMD.

### A.7.2. RESKOOPNET WITH DIFFERENT HYPERPARAMETERS

We test ResKoopNet on the dataset which has 90 initial points, as used in the Section 4.1. The Figure 8 shows that when tunning the number of hidden layers from 1 to 4 and the number of neurons in each layer from 250 to 350, the results change significantly. We can also tell that a suitable NN structure would be about 3 hidden layers with 300 neurons in each hidden layer. More neurons and hidden layers would be fore help, but cost more computational resource. This result also shows that our method is robust to hyperparameters tuning.

## A.8. Hankel-DMD

### A.8.1. JUSTIFICATION OF USING HANKEL-DMD AS COMPARISON IN ALL EXPERIMENTS

Hankel-DMD operates by constructing a Hankel matrix from time-delayed measurements of the system state, based on Takens' embedding theorem, which states that time-delayed coordinates can reconstruct the state space of dynamical systems. Hankel-DMD also falls within the framework of Extended Dynamic Mode Decomposition (EDMD), as it effectively uses time-delayed states as dictionary functions. This connection introduces convergence conditions specific to time-delay embeddings, differing from those associated with standard EDMD implementations. This makes Hankel-DMD a natural choice for comparison in the pendulum system. Specifically, the method enables a more detailed extraction of the system's modes and dynamics, with theoretical guarantees established in works like (Arbabi & Mezic, 2017), which proved its convergence for ergodic systems.

Practically, the approach involves constructing a large matrix of time-shifted copies of measured data, where the number of delays determines how many past states are considered. This theoretically grounded framework is particularly effective when the system states have good temporal resolution and has shown strong performance in analyzing high-dimensional dynamical systems. Consequently, we also apply Hankel-DMD to the turbulence and neural dynamics experiments to evaluate its effectiveness in these representative high-dimensional settings.

As results, although its performance rivals ResKoopNet in the simple pendulum system by showing eigenvalues points near the unit circle and containing some polluted eigenvalues, which are close to the ground truth unit circle, we would like to emphasize that it capture the point spectrum and miss the full spectral information. When it comes to high-dimensional systems, it fails to capture key dynamics in higher-dimensional systems, as seen in the later experiment (Section 4.2 and Section 4.3).

### A.8.2. APPLICATION IN TURBULENCE

Here we present the Koopman modes computed by Hankel-DMD for comparison with the ResKoopNet results. As shown in Figure 9, despite having small residuals, these modes fail to clearly capture the fundamental pressure field structure that was successfully identified by ResKoopNet's first Koopman mode (see Figure 5). The discrepancy observed for

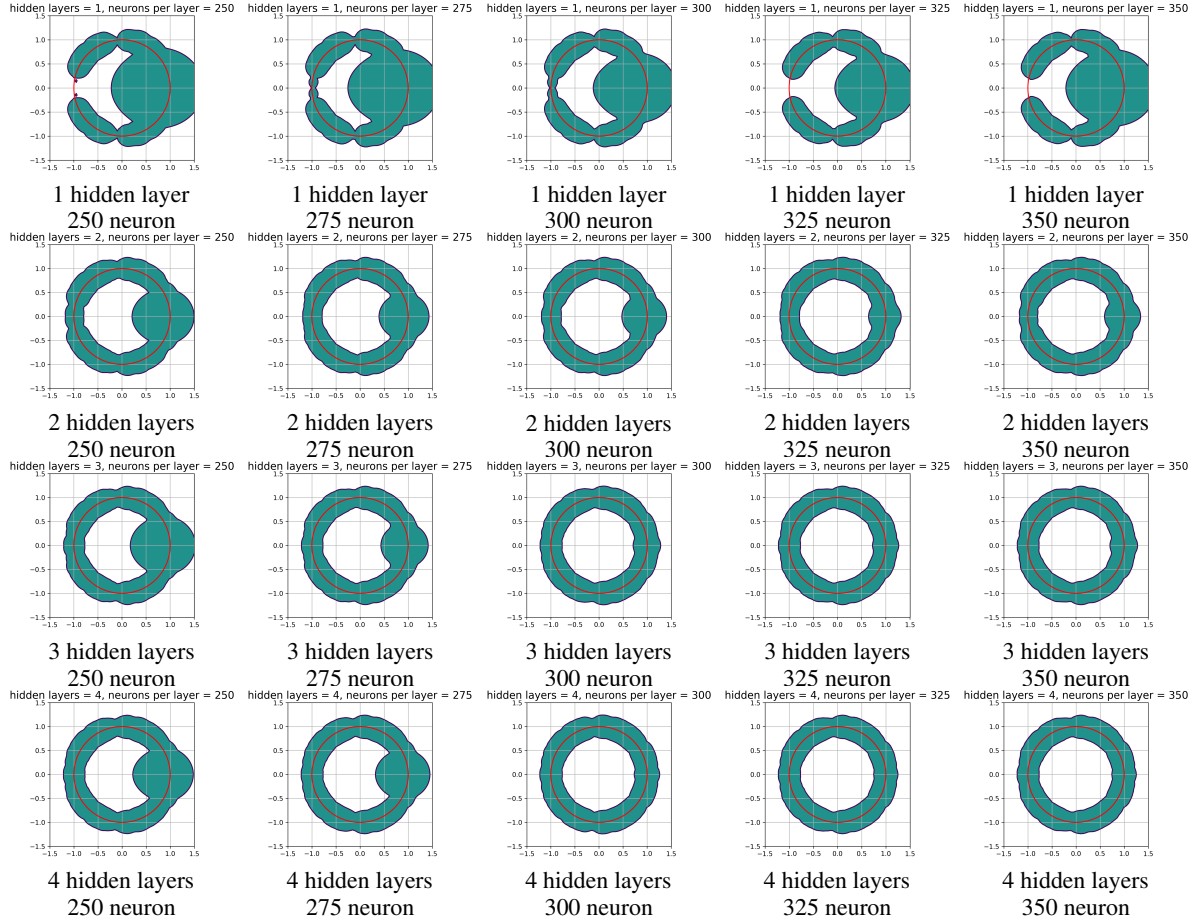

Figure 8: Comparisons of neural network architectures. Each row corresponds to the number of hidden layers (from 1 to 4), and each column shows the model with a different neuron count (250, 275, 300, 325, 350).

Hankel-DMD (Arbabi & Mezic, 2017) is due to its fundamentally different theoretical framework. While ResKoopNet (and other EDMD-based methods) use a Galerkin framework with convergence guarantees established via minimizing the spectral residual, the Hankel-DMD approach is based on Takens' embedding theorem (Takens, 2006). Therefore, the *spectral residual* metric, which is derived and justified within the Galerkin projection framework, does not fully extend to the Hankel-DMD setting. Even if Hankel-DMD yields small residual, this does not ensure that its modes accurately capture the underlying dynamics (such as the fundamental pressure field structure).

### A.9. Practical details for neural data analysis

#### A.9.1. DATASET DETAILS AND EXPERIMENTAL SETUP

The dataset utilized in this study is part of the open dataset provided for the 'Sensorium 2023' competition (Turishcheva et al., 2024b). The dataset consists of calcium imaging recordings from the primary visual cortex of mice. During the experiments, the mice were presented with natural video stimuli while the activity of thousands of neurons was recorded. The objective of the competition is to predict large-scale neuronal population activity in response to different frames of the stimulus videos, based on the hypothesis that population dynamics in the primary visual cortex, driven by visual stimuli, encode significant information about the dynamics of the videos (Basole et al., 2003; Onat et al., 2011; Hénaff et al., 2021).

#### A.9.2. TASK DEFINITION AND RATIONALE

In contrast to the competition's prediction objective, our study focuses on the task of state partitioning of neural signals. While prediction remains feasible, we aim to demonstrate that state partitioning is sufficient to highlight the superiority of ResKoopNet over a series of other methods in uncovering the latent dynamics of the system. Specifically, in each experiment,

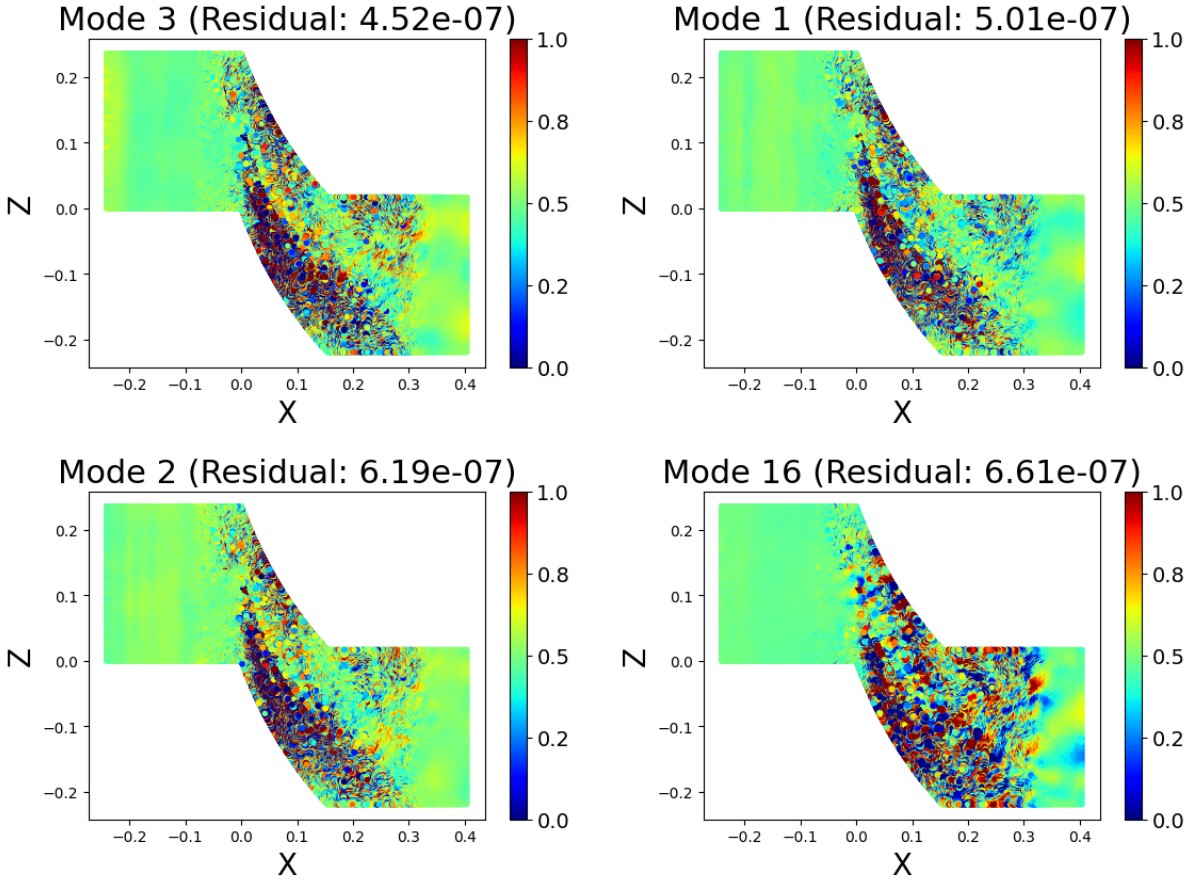

Figure 9: The plots illustrate turbulence detection using the four Koopman modes computed by Hankel-DMD, which are ranked with their corresponding *spectral residual* from the smallest, similar as in ResKoopNet case.

a set of six video stimuli was repeatedly presented to each mouse, creating ideal conditions for defining brain states. The recording setup remained consistent for each mouse, ensuring that the neural activities could be interpreted as originating from the same dynamical system, with the primary variable being the input stimulus.

We hypothesize that during repeated trials with identical visual stimuli, the underlying dynamics of the neural system remain consistent. Consequently, the recurrence of the same brain state is expected during these trials. This provides a reliable basis for testing the efficacy of Koopman decomposition methods in uncovering latent dynamics and distinguishing these states.

### A.9.3. DATASET STRUCTURE AND DIMENSIONALITY

The dataset includes neural recordings from five mice, with each mouse responding to six distinct video stimuli, presented in 9-10 repeated trials (resulting in approximately 60 trials in total). Each trial involves recordings of over 7000 neurons. The duration of each video stimulus is 10 seconds, with a sampling rate of 50 Hz, yielding 300 data points (299 snapshots) per trial. Thus, the data to be analyzed consists of a high-dimensional time series with 7000+ observables per snapshot.

### A.9.4. IMPLEMENTATIONS OF RESKOOPNET AND OTHER CLASSICAL METHODS

We compare here four methods: the proposed ResKoopNet and three classical Koopman decomposition methods for high-dimensional systems: the Hankel-DMD, the EDMD with RBF basis, and the Kernel ResDMD. We applied them to the 5 datasets, although with slightly different implementations and different dimensions of approximated Koopman invariance subspace.

For ResKoopNet, we train the dictionaries with all the snapshots recorded in each mouse such that the total snapshot

number is the product of the snapshot number in one trial and the number of all trials. This is to avoid overfitting with the small snapshot numbers within a trial. The high-dimensional data is first reduced to 300 dimensions with Singular Value Decomposition. The dimension of the Koopman subspace is chosen to be 50 (to be compared with Hankel DMD fairly), consisting of 25 trained bases and 25 pre-chosen ones (constant and the first-degree polynomials of the SVD-ed 24 dimensions). One can find the decomposed eigenfunctions in Figure 6A(top), with a marker of the ground truth state separations based on stimulus identity.

For Hankel-DMD, the Koopman eigenfunctions were approximated using the eigenvectors of the Hankel matrix. Specifically, the Hankel matrix was formed as in Equation 53 from Arbabi & Mezic (2017), using all the observables from one trial of each mouse with a delay of 50. Consequently, the snapshot size became 249 times the observable number, and the resulting number of eigenfunctions was 50, each with a length of 50. The Hankel-DMD eigenfunctions for each trial of data are shown in Figure 6A (bottom), alongside the ground truth trial identities for comparison.

For EDMD with RBF basis, the high-dimensional dataset is first reduced to 300 dimensions with SVD. Then RBF basis is calculated with 1000 RBF functions. The choice of the basis number is decided based on classical experiments of using RBF basis to estimate the Koopman operator of Duffing systems (Li et al., 2017).

For Kernel ResDMD, as it is a variant of Kernel EDMD (Kevrekidis et al., 2016), the dimension of the Koopman invariant subspace should corresponds to the sample number (in time). Given the data size to be 300, we have 299 snapshots, resulting in 299 Koopman bases. The detailed calculated is performed for each trial with the program provided in the original ResDMD paper (Colbrook et al., 2023; Colbrook & Townsend, 2024). We chose the kernel function as the commonly-used normalized Gaussian function in the calculation.

The Koopman eigenfunctions from both ResKoopNet and other methods represent dynamical features corresponding to one of the six video stimuli. To evaluate how well the eigenfunctions capture the latent dynamics, we assess the similarity of the features for trials with the same stimulus and their dissimilarity from those corresponding to different stimuli. Effectively, this makes the problem a clustering task, where the separability of the Koopman eigenfunctions reflects how well they capture the key dynamic components related to the stimuli.

### A.10. Choice justification of dictionary sizes

In this section, we provide justifications for the use of different dictionary sizes (i.e., the number of Koopman eigenfunctions) in the aforementioned four methods for the neural dynamics experiment.

First, the high-dimensional data was pre-processed using SVD to reduce its dimensionality to 300. Then, for the four methods:

1. For ResKoopNet, we selected 25 trained basis functions and 24 first-order monomial basis functions as the dictionary for the 24 reduced observables. This choice ensures the dictionary size is the same as Hankel DMD, making it a fair comparison between the performance of these two methods. The last two methods uses higher dictionary sizes but still remain ineffective in capturing temporal dynamics, thus not affecting the fairness of choosing 50 dictionaries here. The size of the trained dictionary was set to be at least equal to the original observable size to ensure a sufficiently rich dictionary for the Koopman invariant subspace.

2. For Hankel DMD, the number of delays (as dictionary size/number of eigenfunctions) is first constrained by the temporal sample size (i.e., snapshot size) because it cannot exceed the maximum snapshot size. Therefore, it is impossible to choose the same dictionary size as the ResKoopNet example. Choosing the delay too small will result in an insufficient dictionary size to span the Koopman invariant subspace, and too large will reduce the actual snapshot size to estimate the covariance matrices in the estimation of the Koopman matrix. Therefore, we chose a compromise delay number of 50 that satisfies both needs.

3. For RBF basis, in principle, we can use the same dictionary size. However, our previous experience with a similar dataset and the results of using the RBF basis for the EDMD method all suggest that the performance will be better with more dictionary functions. Therefore, we chose 1000 RBF basis and the original 300 first-order monomial basis as a better condition compared to the same dictionary size with ResKoopNet.

4. For Kernel ResDMD, the dictionary size is theoretically determined to be the number of snapshots. Therefore, we cannot make the dictionary size consistent with the ResKoopNet example.

Based on the above justifications, we believe our choices of dictionary sizes are reasonable and ensure a fair comparison across the methods.

### A.10.1. VISUALIZATION AND CLUSTERING PERFORMANCE

To visualize the clustering of high-dimensional Koopman eigenfunctions, we perform dimensionality reduction using Multi-dimensional Scaling (MDS). MDS is particularly useful for visualizing high-dimensional data by preserving pairwise similarities (Kruskal, 1964) (here we use correlation as a measure of similarities). While UMAP (McInnes et al., 2018) and t-SNE (Van der Maaten & Hinton, 2008) are alternative visualization methods, with different emphasis on global-local relationships, we primarily use MDS in this study and provide UMAP and t-SNE results in the supplementary materials (see Appendix Figure 13A, B, Appendix Figure 14C, D and Appendix Figure 15C, D). UMAP in implementation is still correlation-based. For t-SNE estimation we use the perplexity of 15, as a value for optimal separation.

By applying MDS, the high-dimensional eigenfunction-based features are reduced to a low-dimensional space. For illustration, we present the results of reducing the feature space to two dimensions (Figure 6B-E). The ResKoopNet reduced features for the six types of trials (corresponding to the six video stimuli) are well-separated for all five mice (Figure 6B). In contrast, the Hankel-DMD features show no clear clustering structure (Figure 6C). Similarly, the features produced by EDMD with an RBF basis and Kernel ResDMD do not show clear separability (Figure 6D-E, Appendix Figure 14B-D, Appendix Figure 15B-D).

### A.10.2. CLUSTERING QUALITY METRICS

We further quantified the clustering quality by calculating the Davies-Bouldin Index (DBI) for both Koopman decomposition methods across all mice (Figure 6F). The DBI is designed to assess the compactness of clusters and the separability between them. A lower DBI indicates better clustering performance. ResKoopNet features yield significantly lower DBI scores compared to other methods, confirming that ResKoopNet produces more clearly defined clusters corresponding to the ground truth trials. Similar clustering results are observed with UMAP and t-SNE (see Appendix Figure 16), further supporting the superior performance of ResKoopNet in capturing the latent dynamic structure compared to the other classical methods.

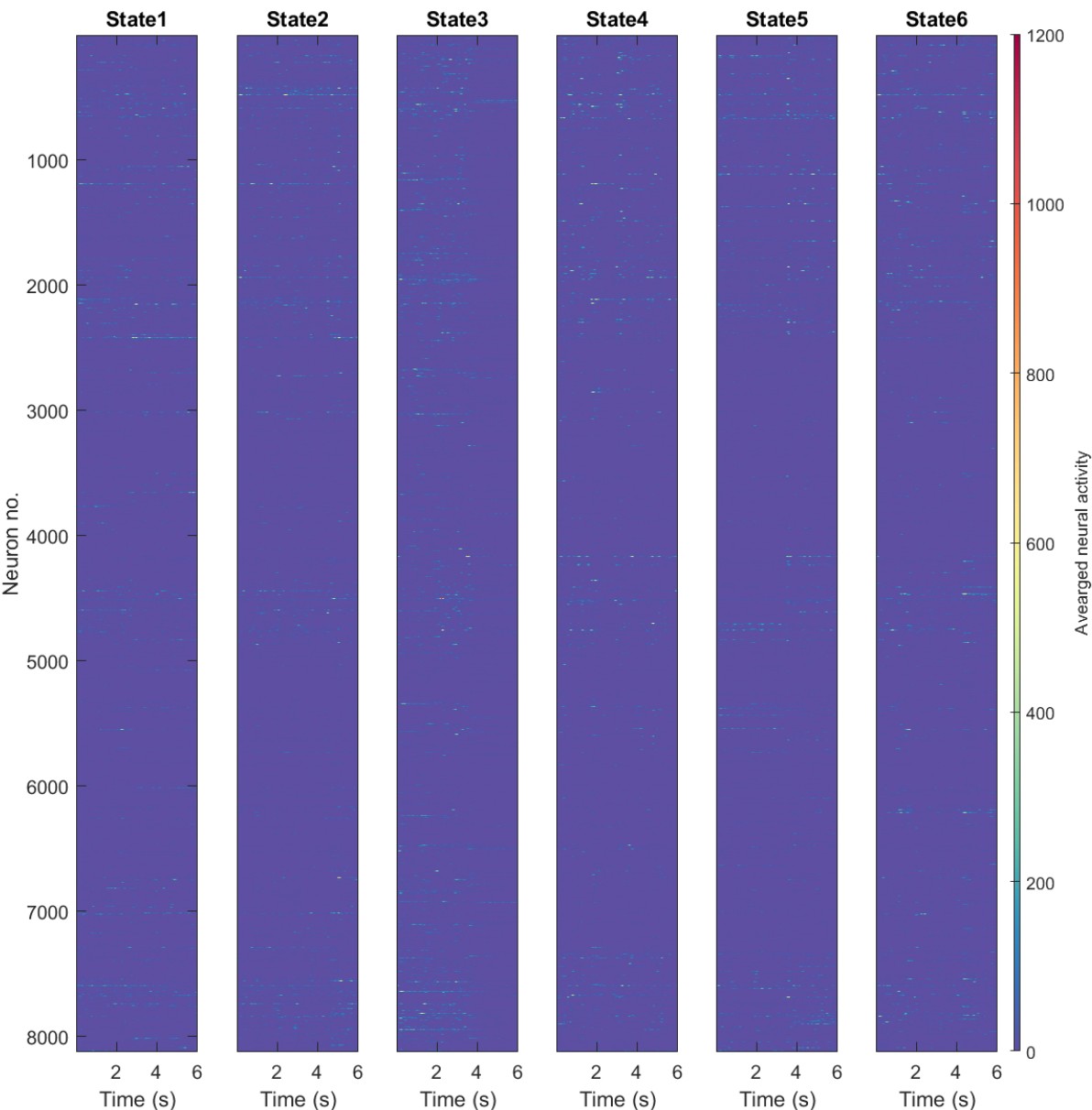

Figure 10: Trial-averaged neural responses to 6 video stimuli (marked by 6 states).

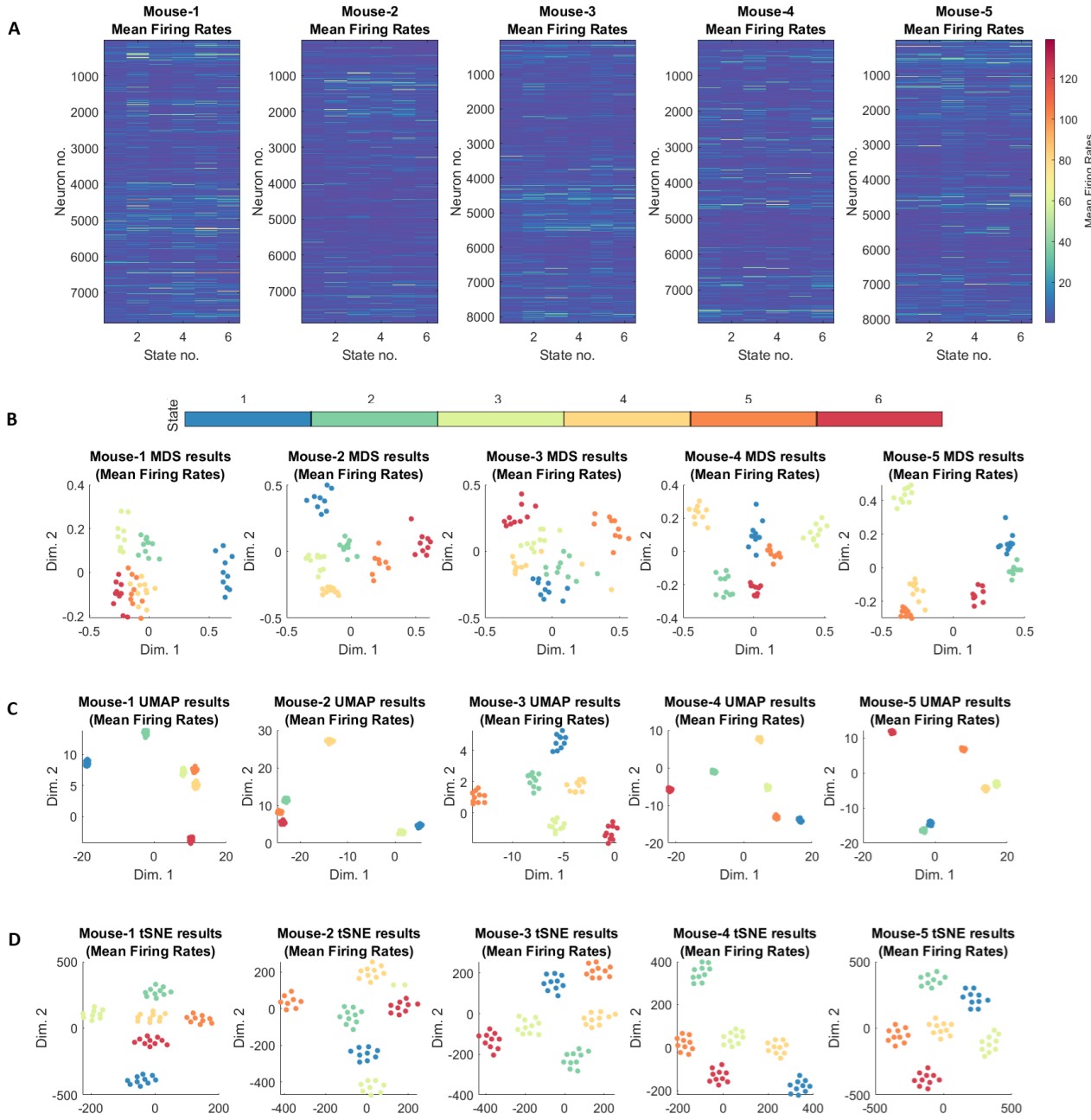

Figure 11: (A) Mean firing rates (calcium spike) of each mouse in response to 6 video stimuli (marked as states). (B) Clustering performance based on the mean firing rates of each mouse to 6 video stimuli, visualized with Multi-dimensional Scaling (MDS). (C) Same as (B) but visualized with UMAP. (D) Same as (B) but visualized with t-SNE

.

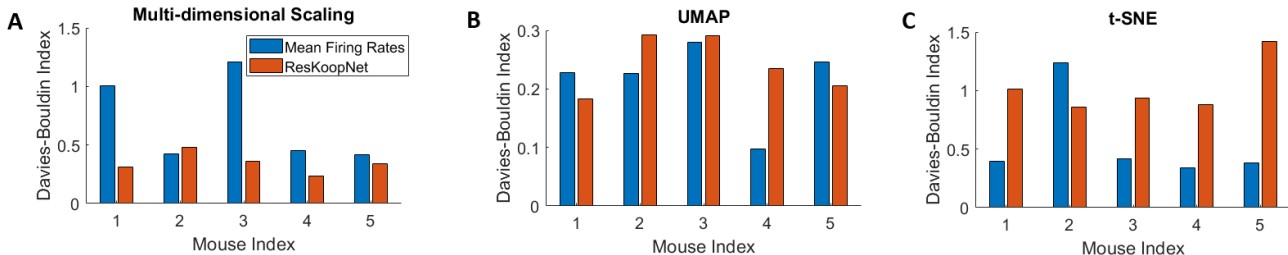

Figure 12: Comparison of clustering quality based on Davies-Bouldin Indices for clustering with mean firing rates and Koopman eigenfunctions learned from ResKoopNet, visualized in 2D space with Multi-dimensional Scaling (A), UMAP (B) and t-SNE (C).

Figure 13: State Partition performance with eigenfunctions for ResKoopNet (50 bases) and Hankel-DMD in 2D space visualized with UMAP (A) and t-SNE (B).

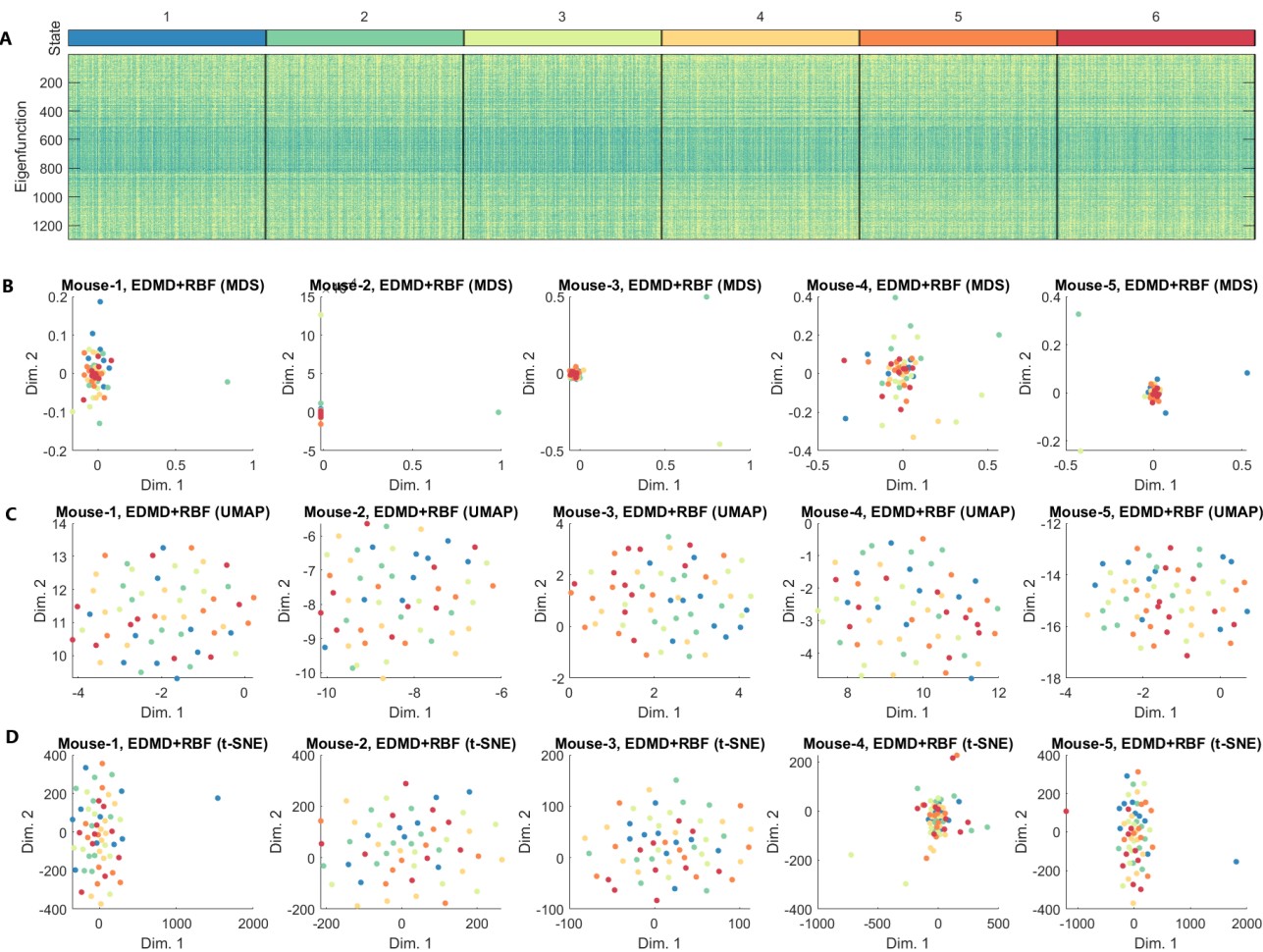

Figure 14: Full results of EDMD with RBF basis. (A) 1301 Koopman eigenfunctions estimated by EDMD with RBF basis in 6 states characterized by 6 different video stimuli in an example mouse. Eigenfunctions in each trial of each state contain 300 data points (10s with a sampling rate of 50Hz). (B) 2-D representation of Koopman eigenfunctions for each trial of all tested mice, calculated by EDMD with RBF basis and reduced by Multidimensional Scaling (MDS). No clear separation of states can be seen from the reduced representation. (C) Same as (B) but visualized with UMAP. No clear separation of states can be seen from the reduced representation. (D) Same as (C) but visualized with t-SNE. No clear separation of states can be seen from the reduced representation.

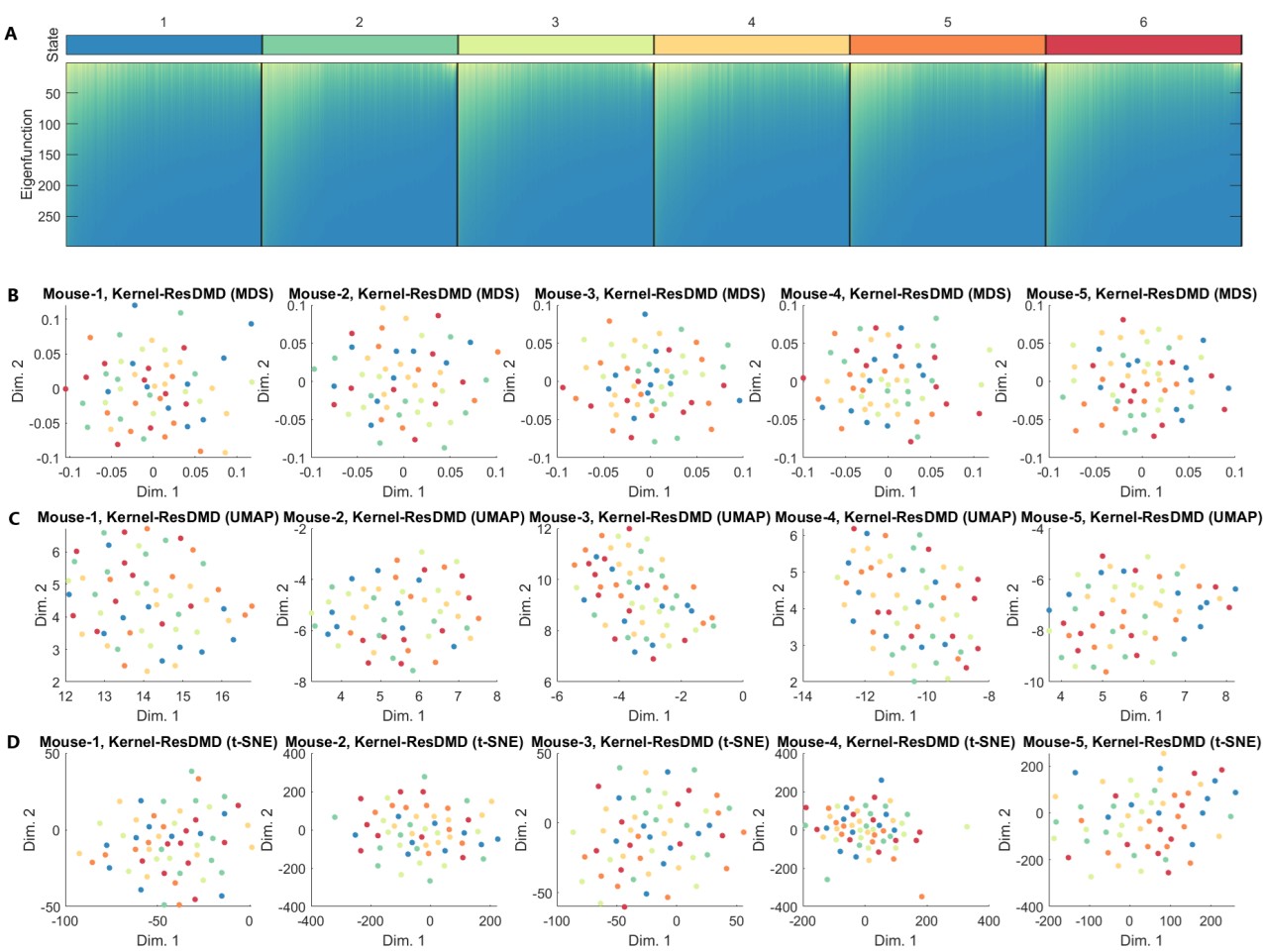

Figure 15: Same as Figure 14 but estimated with Kernel ResDMD, with 299 basis of the Koopman subspace, thus 299 eigenfunctions.

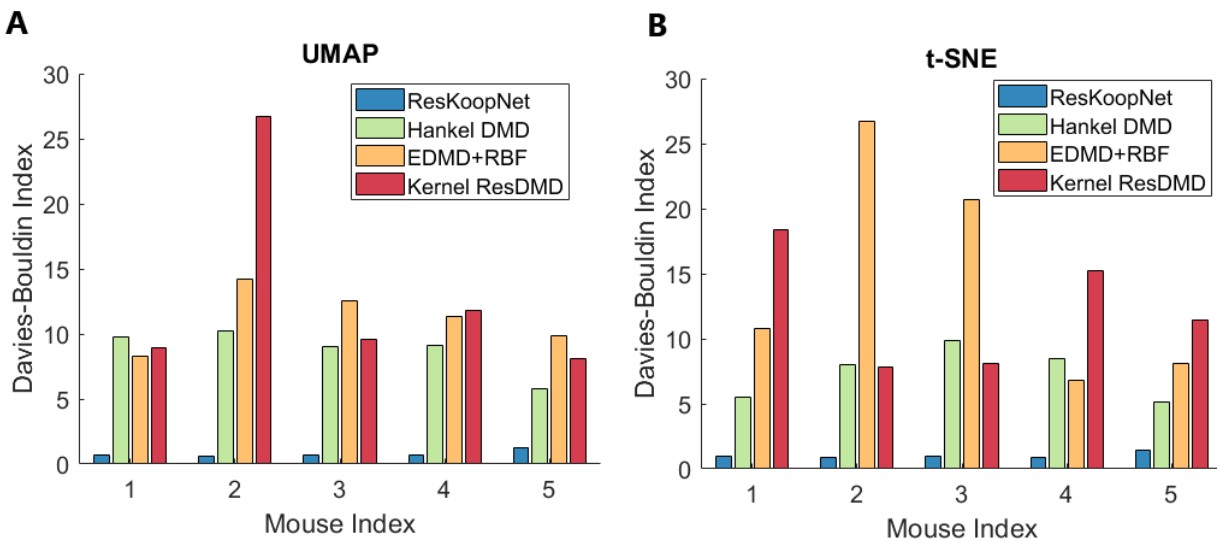

Figure 16: Davies-Bouldin Indices evaluating the clustering performance of dynamical components learned by four methods (ResKoopNet(50 bases), Hankel DMD, EDMD+RBF, and Kernel ResDMD) across five mice. Comparisons are shown using UMAP (A) and t-SNE (B).

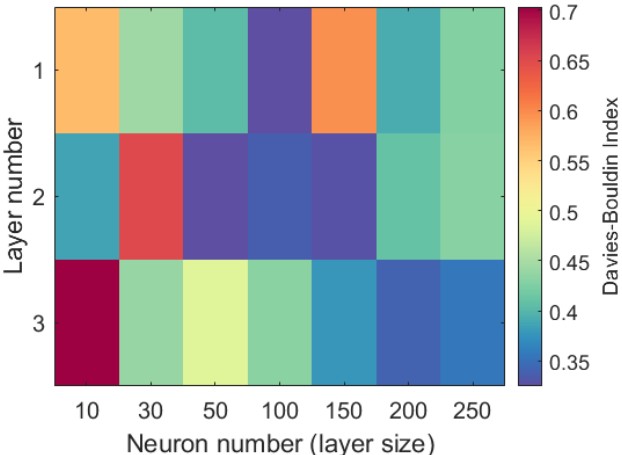

Figure 17: Davies-Bouldin Indices for different neural network hyperparameters in training ResKoopNet with 50 bases in an example mouse. Note that lower values indicate better clustering performance. With small layer size and layer number, the clustering performance is not stable but becomes robust when the layer size reached 200 and layer number reaches 3.

