# OpenReview forum: "ResKoopNet: Learning Koopman Representations for Complex Dynamics with Spectral Residuals"
_ICML.cc/2025/Conference — ICML 2025 poster_

### Official Review · Reviewer_2cVC · 2025-03-13

**Overall Recommendation:** 3

**Summary:**

This paper introduces a new deep learning based approach for approximating Koopman eigenvalues based on Residual DMD. By learning a dictionary representation that minimizes a spectral residual loss function, the authors demonstrate their approach is able to perform well on several data sets.

**Claims And Evidence:**

All major claims are supported by clear evidence.

**Essential References Not Discussed:**

The authors cited everything that I think is essential (but could use with more referencing of those works throughout the text, as noted above).

**Experimental Designs Or Analyses:**

The experiments were largely well done, but I have several specific comments/questions:

1. The authors claim that ResDMD requires 964 dictionary elements to compute the pseudospectra of the nonlinear pendulum (Sec. 4.1). However, Fig. 1 of Colbrook and Townsend (2024) only shows the spectra computed for 150 and  964 dictionary elements. The authors show that 300 dictionary elements does not lead to the complete pseudospectra (Fig. 4), but they do not show that at least 964 are needed. It could be that significantly fewer dictionary elements than 964 are needed (but greater than 300). I think therefore this claim should be modified.

2. The authors show that the higher Koopman modes computed by ResKoopNet approximate acoustics in the turbulence example. Figs. 5c and d look identical to me. Could this be an error and the same figure plotted twice? If not, is there a reason the 6th and 7th Koopman modes are so similar?

3. The results on using Koopman mode decomposition for clustering neural activity across stimuli is interesting. I think there are three basic analyses that could strengthen these results. First, could the authors plot the neural population recordings (averaged across trials) for each stimuli. Is there a clear difference in the activity across stimuli (in which case, maybe such a dynamics based clustering is not necessary). Second, what happens if you cluster the trials just based on the mean activity (for each neuron) across the trial? I could imagine different visual stimuli driving different average activity in the neural population, that could be picked up by the Koopman eigenfunctions associated with eigenvalue 1. And third, what happens if you use fewer Koopman modes, computed from ResKoopNet, in your clustering? It seems like comparing having 500 modes vs. having 50 (in the case of Hankel-DMD) could affect the comparison.

**Methods And Evaluation Criteria:**

The authors evaluate their computational framework on two benchmark data sets (pendulum and turbulence) and one data set that has not been frequently studied (mouse neural activity). The authors compare their approach to several different methods that are commonly used, and appear to make the comparisons fairly.

**Other Comments Or Suggestions:**

The authors rely a number of times on referencing results from the original ResDMD paper. This makes it harder for the reader who is not very familiar with ResDMD to follow at places, and to compare the results. I think one last way for the authors to improve their paper is to include more results/discussion to make the paper as self contained as possible, even if this all this additional information is added to the Appendices.

**Other Strengths And Weaknesses:**

**Strengths**
1. I thought the idea of using the residual spectral loss was innovate and appeared to lead to strong results.
2. The application of ResKoopNet to neural data (while I have some reservations about it, see above) is novel and interesting.
3. Developing better tools for approximating the continuous part of the Koopman spectra is important and I think a contribution of this paper (that could even be discussed more in the intro).

**Weaknesses**
1. The experimental results could be improved, as discussed above.
2. The authors could make their contributions more clear (and not make claims that could be misleading about their contributions), as discussed above.
3. The authors could make the paper easier to follow by including more discussion on ResDMD, particularly on Kernel ResDMD, since they compare to this method several times. Additionally, writing out the Koopman mode decomposition could allow the authors to introduce Koopman methods better. Finally, explaining why the true pseudospectra covers the entire unit circle in Sec. 4.1 would be helpful.
4. Finally, there are a number of typos and grammatical errors throughout the paper. While I very much understand the presence of these things, the amount was bordering on making it harder to read.

**Questions For Authors:**

1. Are Fig. 5c and d the same figure? If not, why are the modes so similar?
2. How much do the clustering results in Fig. 6 rely on the learned dynamics (as opposed to the trial average responses of the neurons/mean firing rate for each stimuli)? How dependent are they on the exact number of Koopman eigenfunctions used for ResKoopNet?

**Relation To Broader Scientific Literature:**

This work could be improved by motivating more clearly why ResDMD is such a powerful method and why using the spectral residual loss function is so important. Why not use, for instance, the dictionary learning EDMD to find a good spectral approximation. I think the authors have reasons for this sprinkled throughout the paper, but making it more clear would be helpful. Additionally, the authors discuss dictionary learning EDMD and work showing convergence bounds of EDMD (Korda and Mezic, 2018), but then they also say things like "ResKoopNet employs neural networks to optimize dictionary functions" (page 1) and "EDMD lacks theoretical gaurantees of convergence" (page 1), without discussing this highly relevant work. Finally, the authors say that "as shown in (6), ResKoopNet provides an explicit expression for $\tilde{K}$" (page 3). But this expression is just from EDMD, and not specific to ResKoopNet. These last two points give slightly misleading representations on what this works contributions are.

**Theoretical Claims:**

I did not follow the discussion of Barron spaces and convergence in Appendix A.3. This is not my expertise, but I did feel that the introduction to Barron spaces was a little terse and it was not clear to me why one would expect them to be appropriate spaces to consider for ResKoopNet, since the networks used were three layers (Sec. 3.2 -- not two layers, as mentioned in Appendix A.3).

---

> ### Author Rebuttal · Authors · 2025-04-01
>
> 1. Thank you for your insightful suggestions. The updated figures and proof can be seen here: https://anonymous.4open.science/r/rebuttal_materials-14918/
>
> 2. Thank you for raising this point about Barron spaces and the network architecture. To clarify, Barron spaces (Appendix A.3) provide a theoretical framework for analyzing functions efficiently approximated by neural networks with certain properties. Their mathematical properties, notably being dense in  $L^2$ spaces under certain conditions, ensure that functions in  $L^2$  can be approximated arbitrarily well by Barron functions. This justifies applying Theorem A.3 and supports our extended analysis (see the file **convergence_proof.pdf**).
>
> 3. Regarding the network architecture notation: the “2-layer” network mentioned in the theorem refers to a single hidden layer plus an output layer, which is the standard notion in approximation theory. Practically, we used three hidden layers (Section 3.2) for improved empirical performance, but insights from the 2-layer theory still inform deeper network convergence behavior (see **convergence_proof.pdf**).
>
> 4. The convergence analysis in Section A.4 relies on this framework because it establishes that as the number of neurons increases, the approximation error decreases at a specific rate (proportional to $1/\sqrt{r}$ where $r$ is the number of neurons). This gives us confidence that with sufficiently large networks, ResKoopNet can theoretically approximate the true Koopman eigenfunctions with quantifiable error bounds, which is essential for the spectral analysis guarantees we discuss (see **convergence_proof.pdf**).
>
> 5. You are correct that the original ResDMD requires fewer basis functions. We discuss this further and include a new experiment (**Fig_7.png**). Still, results show that ResDMD needs around 500 manually selected dictionary elements, whereas ResKoopNet achieves better performance with only 300 basis functions without manual selection.
>
>
> 6. Regarding Fig. 5(c) and (d): You are correct; these were identical due to complex conjugate eigenvalues. In the updated version (**Fig_5_new.png**), we've repeated the experiment using 250 dictionary elements (matching ResDMD from Colbrook & Townsend, 2024), resulting in distinct Figures 5(c) and (d). We have also added singular value plots in the new Figures 5(e) and (f).
>
> 7. Regarding the neural experiment, as suggested by the reviewer, we have included new figures:
>
> (1) To ensure a fair comparison with the 50 bases in Hankel-DMD, we re-estimated the Koopman eigenfunction from ResKoopNet using 50 dictionaries (24 SVD truncated bases, one constant, and 25 trainable bases). As a result, the performance is similarly good to the performance with 501 bases. For details of approximated averaged eigenfunctions and clustering results please refer to **Fig_6_new.png, Fig_13.png and Fig_16.png**. A hyperparameter scanning is also included in **Fig_17.png**.
>
> (2) The reviewer is right to point out that both the averaged response across trials and the mean firing rates are sufficiently distinct for each video stimulus. **Fig_10.png** shows trial-averaged neural activities for an example mouse, and **Fig_11.png(A)** presents mean firing rates across all mice for the six stimuli.**Fig_11.png(B–D)** demonstrates clustering directly based on mean firing rates, confirming that mean firing rate information alone is sufficient, with performance comparable to ResKoopNet (**Fig_12.png**).
>
> (3) We agree clustering via Koopman eigenfunctions is biologically not strictly necessary here given the clear stimulus-driven differences. However, this dataset provides an ideal benchmark for Koopman eigenfunction estimation methods. Notably, standard approaches like Hankel DMD, kernel ResDMD, and EDMD (RBF basis) fail to differentiate these distinct processes, likely due to limitations in handling high-dimensional data or eigenfunction estimation.
>
> (4)The effectiveness of ResKoopNet in accurately estimating Koopman eigenfunctions demonstrates its strength and broader applicability, particularly for more challenging tasks such as unsupervised latent brain state identification or decoding/reconstructing object movements from the videos, forming promising directions for future research.
>
> 8. Regarding 'Why not use, for instance, the dictionary learning EDMD to find a good spectral approximation': The Koopman matrix updating formula $\tilde{K}$ (Eq.6) matches EDMD, but it is derived from a distinct loss function (Eq.5), as shown in Appendix A.2. This loss function difference is fundamental; ResDMD (via Galerkin approximation, Eq.(2)) approximates both $\mathcal{K}$ and $\mathcal{K}^*\mathcal{K}$, whereas EDMD approximates only $\mathcal{K}$. Thus, during optimization, ResKoopNet yields a $\tilde{K}$ that better captures Koopman's spectral properties.

---

> > ### Comment · Reviewer_2cVC · 2025-04-07
> >
> > I appreciate the authors thorough response to my questions. The answers have sufficiently addressed my questions/concerns. I am going to maintain my score of a 3, however I feel more confident in that score and more confident in supporting this paper.

---

> > > ### Author Response · Authors · 2025-04-08
> > >
> > > Thank you for your comments and for taking the time to review our paper. I'm glad to hear that our responses helped clarify your concern and increased your confidence in the work.

---

### Official Review · Reviewer_K3yo · 2025-03-13

**Overall Recommendation:** 2

**Summary:**

This research presents a novel method for approximating the spectral components of the Koopman operator for discrete-time deterministic dynamical systems by minimizing spectral residuals. Unlike traditional methods that rely on predefined dictionaries, this approach utilizes a neural network to optimize dictionary functions to discover a more precise and complete spectrum of the Koopman operator for complex dynamical systems. The method employs an alternating optimization procedure, where a neural network learns the suitable observables while the Koopman matrix is iteratively updated via least-squares estimation. The method's effectiveness is validated on a pendulum system, a turbulence model, and neural recordings, demonstrating superior accuracy over classical methods.1

**Claims And Evidence:**

1. The authors claim that their method addresses spectral inclusion and reduces spectral pollution for complex dynamical systems of high dimensions, which is not well-supported. In none of the experiments do they show that there exists a subset of the true spectrum that is not attainable from ResDMD or other methods, but instead that their method can do that. In Experiment 1, which is a low-dimensional system, they only show that for finding the whole spectrum, ResDMD needs a bigger size dictionary. Also, it is noteworthy that Hankel-DMD works almost as well as the proposed method. In Experiment 2, the authors do not directly compare their method with ResDMD using the same size dictionary. While they are using Nk = 300, the figures in the ResDMD are when Nk = 250. This problem also exists in the last experiments, and the size of the dictionaries is very different. So in a nutshell, for the first experiment (low-dim) they did not show that they are able to find some part of the spectrum that the other methods cannot, and for the others (high-dim) the comparison is not fair in my opinion. Thus, I would suggest the authors provide more evidence to support their argument.

2. The authors use the term “optimal” representation several times in the paper, but they do not specify in which sense the neural network's learned representation is optimal. How do they know that? Since the cost function they are using is an estimate of spectral residual, with a finite number of samples, one cannot be sure that if it is small, it means that the real object is also small. Furthermore, there might be noise in the data.

3. In addition, the claim for convergence analysis is a bit shaky, which I will elaborate on in the theoretical claims section.

**Essential References Not Discussed:**

N/A

**Experimental Designs Or Analyses:**

I checked the experiments and analysis. As I mentioned before, the experiment design could be improved to make the comparison more fair.

**Methods And Evaluation Criteria:**

Yes, the proposed method is sensible in general. Using neural networks to learn representations for the Koopman operator is a well-known technique in the field. Also, the datasets used are aligned with the proposed method and the problem. However, I am not sure that the evaluation criteria for showing the superiority of the methods make sense if there is not a fair comparison between different methods.

**Other Comments Or Suggestions:**

Typos:
“ … its only filters precomputed spectra …” → “... it only filters …”
"... These results are presented in Appendix Figure 7 and Appenxic A.7.2." → appendix A.7.2
“In this chapter ..” → “In this section…”

**Other Strengths And Weaknesses:**

The paper is highly original in its integration of spectral residual minimization with neural network-based dictionary learning, addressing key limitations of existing Koopman approximation methods. Its significance lies in improving spectral accuracy for complex dynamical systems, with strong implications for physics, engineering, and neuroscience. In terms of clarity, the paper is generally well-written and the concepts are explained in a logical flow. Overall, the paper presents an important contribution but would benefit from broader benchmarking and a clearer discussion of trade-offs.

**Questions For Authors:**

1. The authors only compared their method with classical methods, which use kernel functions. I wonder if there are any DMD methods leveraging neural networks that could be included for comparison?
2. It is unclear how the authors determined the hyperparameters for the proposed method and the other methods. I wonder if hyperparameter tuning was performed, especially since some of the comparison methods are kernel methods, where using different kernel functions might yield different results. If yes, I think it will be great to mention to it in the main body.
3. In the turbulence experiment, the color maps vary across different Koopman modes and methods. Would it be possible to use normalized Koopman modes to better facilitate comparison and verification of the results?
4. The authors reported that despite having small residuals, the Hankel-DMD modes fail to clearly capture the fundamental pressure field structure. Does this not suggest that the residual metric may not be a reliable indicator of representational quality, and thus the final representation may not be truly “optimal”?

**Relation To Broader Scientific Literature:**

The key contributions of the paper build upon prior advancements in Koopman operator theory, particularly addressing limitations in existing spectral approximation methods such as Extended Dynamic Mode Decomposition (EDMD) (Williams et al., 2015) and Residual Dynamic Mode Decomposition (ResDMD) (Colbrook & Townsend, 2024). While EDMD provides a finite-dimensional approximation of the Koopman operator, it suffers from spectral pollution and lacks guarantees of capturing the full spectrum, especially in high-dimensional systems. ResDMD improves upon this by introducing spectral residuals to filter inaccurate eigenvalues, but it still relies on predefined dictionaries and does not refine spectral estimates. ResKoopNet advances this line of research by optimizing dictionary functions through a neural network, thus addressing the spectral inclusion problem and improving spectral accuracy. This aligns with recent trends in machine learning-assisted dynamical systems analysis, such as deep autoencoders for Koopman embeddings (Lusch et al., 2018) and kernel-based Koopman methods (Kevrekidis et al., 2016), but differs in its explicit spectral residual minimization approach.

**Theoretical Claims:**

Yes, I checked all the theoretical claims in the appendix. The authors demonstrate equivalences between different equations presented in the main body of the paper, which appear correct. However, the convergence analysis is, in my opinion, somewhat shaky. First, the proposed method uses a three-layer neural network, whereas the theory provided is for two-layer networks. Second, the norm of the functions derived from density arguments is unknown. Thus, as we considering sum of the norm of these functions for each mode it is not clear how one might ensure the epsilon guarantee.

---

> ### Author Rebuttal · Authors · 2025-04-01
>
> 1. We would like to thank the reviewer for the helpful comments. The updated figures and proof can be seen here: https://anonymous.4open.science/r/rebuttal_materials-14918/
>
> 2. Regarding “Hankel-DMD works almost as well as the proposed method”: While Hankel-DMD performs well in Experiment 1's lower-dimensional case, it underperforms in Experiments 2 and 3's higher-dimensional settings. We included it across all experiments for consistency and fair comparison. In Experiment 2, ResKoopNet successfully captured a dominant Koopman mode showing the main air pressure structure near the airfoil that both ResDMD and Hankel-DMD missed (see ResDMD Fig.17 and our Appendix Fig.7) In Experiment 3, ResKoopNet succeeded whereas Hankel-DMD and others failed. These results show that our method addresses the spectral inclusion by capturing important spectral components that other methods miss.
>
> 3. We acknowledge that our original turbulence example used 300 basis functions, while ResDMD used 250. We re-did the experiment and the updated results are shown in **Fig_5_new.png**.
>
> 4. In the neural experiment, to ensure a fair comparison with the 50 bases in Hankel-DMD, we re-estimated Koopman eigenfunctions from ResKoopNet using 50 dictionary elements (24 SVD-truncated bases, one constant, and 25 trainable bases). The resulting performance is comparably strong to the original 501-basis case. Detailed approximations of eigenfunctions and clustering outcomes are provided in the updated **Fig_6_new.png, Fig_13.png and Fig_16.png**.
>
> 5. Regarding the optimality comments:
>
> (1) Regarding the comment that ‘in which sense the neural network's learned representation is optimal’: the ‘optimal’ representation in ResKoopNet is not optimal in the classical sense (e.g. a unique minimizer of an L2 error as in EDMD); rather, it means that the learned dictionary minimizes a very specific loss: ‘spectral residual’ so that the finite-dimensional Koopman operator best captures the true spectral properties (both discrete and continuous as explained in ResDMD of the underlying Koopman operator given the data. In other words, by minimizing the spectral residual, the network is trained to produce a dictionary (or representation) that ‘optimally’ balances the ability to approximate the Koopman eigenpairs against the error introduced by discretization and finite sampling. Minimizing this loss(Eq.7 in the paper) pushes the network toward a representation where the computed eigenpairs have a very low spectral residual.
>
> (2) Regarding the comment that 'with a finite number of samples, one cannot be sure that if it is small, it means that the real object is also small', the authors justify the 'optimality' by proving convergence results in Appendix A.4(see **convergence_proof.pdf**): as the number of samples increases and as spectral residual loss function $J(\theta)$ below a threshold, the approximation of the dictionary converges (and hence the `optimality’ hold). So while practically the cost function is an estimate, the theory from ResDMD assures that, in the limit, a near-zero cost implies a good approximation of the true Koopman spectrum.
>
> (3) Additionally, the convergence analysis in Appendix A.4 (pages 14–15, in red) is expanded to demonstrate the convergence of both the dictionary and the spectrum, see **convergence_proof.pdf**.
>
> (4) The data may contain noise, but our work uses a deterministic framework without accounting for stochastic effects. We plan to address noise and stochasticity in future research (see the Conclusion Section (second paragraph)).
>
> 6. Regarding other DMD methods with neural networks: Since ResDMD is EDMD-based, we compared it with EDMD-DL(Li et al., 2017). ResDMD and kernel-ResDMD have full spectrum theoretical guarantees but do not address spectral inclusion issues, while EDMD and EDMD-DL are effective in both low and high dimensions, they lack similar guarantees as ResDMD. Future work will explore advanced neural network structures, e.g., PINO or KAN, and include further comparison with other neural network-based DMD methods.
>
> 7. We performed hyperparameter tuning and included them in **Fig_8.png**, demonstrating the robustness of our parameter choices. The hyperparameter scanning results for the neural experiment are included in **Fig_17.png**: with smaller layer size and layer number, the clustering performance is not stable but becomes robust after the two hyperparameters reach a threshold. Therefore we have chosen 3 layers of 200 neurons.
>
> 8. We have now used the same normalized color map value for each Koopman mode figure (**Fig_5_new.png**).
> Hankel-DMD does not perform as well as ResKoopNet in Fig. 7 of Appendix A7.2 because its foundation(Takens’ embedding) is different from the Galerkin framework used in EDMD and ResDMD, from which the spectral residual metric is derived. Thus, even a small residual for Hankel-DMD may not capture key dynamics (like the main pressure field structure and clustering brain state).

---

### Official Review · Reviewer_QgoQ · 2025-03-15

**Overall Recommendation:** 3

**Summary:**

The paper focuses on Koopman operator analysis and builds on the Residual Dynamic Mode Decomposition (ResDMD), which uses the spectral residual to evaluat the accuracy of a Koopman operator approximation and to perform filtering of a computed spectrum. Here, the proposal is to use the spectral residual iteratively during the spectral estimation process, iterating between optimizing a neural network that parameterizes a dictionary and minimizing the residual to identify the Koopman eigenpairs. The paper reports numerical experiments for multiple systems/datasets including a pendulum system, a turbulence system, and neural recordings from the visual cortex of mice.

**Claims And Evidence:**

Please see "Other Strengths and Weaknesses"

**Essential References Not Discussed:**

[R1] Colbrook, Matthew J. "Another look at Residual Dynamic Mode Decomposition in the regime of fewer Snapshots than Dictionary Size." Physica D: Nonlinear Phenomena 469 (2024): 134341.

**Experimental Designs Or Analyses:**

Please see "Other Strengths and Weaknesses"

**Methods And Evaluation Criteria:**

Please see "Other Strengths and Weaknesses"

**Other Comments Or Suggestions:**

The paper’s contribution would be considerably stronger if it:
(a)	Expanded on the convergence analysis in Appendix A.3. The current analysis is very brief and does not provide a clearly-stated convergence result. The final discussion essentially focuses on the existence of the dictionary rather than the ability of the algorithm to converge to it.

(b)	Conducted a more thorough investigation of the potential ways to incorporate the neural network (architectural choices, learning behaviour, convergence). Since this is the major novel contribution of the work, seems neglected in the experimental analysis.  I suspect that the authors were unaware of [R1] (or even conducted some of the research associated with this work prior to the emergence of that paper). If this paper were the first to propose minimizing the spectral residual (as well as incorporating a neural network) then the contribution would be more substantial and it would be more reasonable to focus entirely on other non-neural experimental aspects.

{R1] Colbrook, Matthew J. "Another look at Residual Dynamic Mode Decomposition in the regime of fewer Snapshots than Dictionary Size." Physica D: Nonlinear Phenomena 469 (2024): 134341.

**Other Strengths And Weaknesses:**

Strengths
S1.	The paper introduces a novel Koopman analysis framework.

S2.	The reported experiments are thorough and the results demonstrate a noticeable improvement over existing techniques.

S3.	Section 4.3 is particularly interesting and highlights the strengths of the proposed methods in terms of its ability to capture key dynamics and perform effective clustering.

S4. The paper provides a good discussion of relationships to other neural-network based Koopman operator frameworks (Appendix A.4).

S5. The discussion of the computational costs (Appendix A.5) is helpful.

S6. There is good detail and justification for experimental design choices (Appendix A.9)

S7. The visualizations and clustering quality analysis, and related discussion, provide valuable insight into the behaviour of the proposed method (Appendix A.9).

Weaknesses

W1.	The paper’s novel technical contribution is limited. The paper does not cite [R1] (available from 30. Aug 2024), which outlines an algorithm (Algorithms 7 and 8) to compute the pseudospectra by minimizing the residual. This appears to be identical to the minimization proposed here for a fixed dictionary. Given that this component of the algorithm has already been published, the remaining innovation of this paper consists of incorporating a neural network to parameterize the basis functions and iterating between the minimization steps. As the authors acknowledge, this approach has been widely used in the neural Koopman operator literature, albeit with the focus on the prediction-based loss function (e.g., Li et al., 2017).

W2.	Despite its key innovation being the introduction of a neural network to jointly learn the dictionary while minimizing the spectral residuals to determine the Koopman eigenpairs, the experimental section of the paper does not explore any design choices related to the neural network (architecture, layers, convergence, etc.)

[R1] Colbrook, Matthew J. "Another look at Residual Dynamic Mode Decomposition in the regime of fewer Snapshots than Dictionary Size." Physica D: Nonlinear Phenomena 469 (2024): 134341.


AFTER REBUTTAL:
(1) The authors have clarified the differences between their work and [R1].
(2) There has been further experimental exploration of the architectural choices.
(3) The authors have provided further theoretical results concerning convergence. While these are welcome and strengthen the paper, I would observe that the inclusion of a proof in the rebuttal appears to violate the review process policies, which impose a strict character limit and require that links only contain figures and tables. This inclusion potentially provides the authors of this paper with an unfair advantage. I would like the Area Chair to comment on this.

With the above factors in mind, I have increased my overall score from 2 (weak reject) to 3 (weak accept).

**Questions For Authors:**

Q1. It is possible that there is a more meaningful distinction between Algorithms 7 & 8 in [R1] and the procedure proposed here, and that I have misinterpreted one of the algorithms due to notational differences. If so, I would appreciate if the authors could clarify what they perceive as the critical difference.

**Relation To Broader Scientific Literature:**

Please see "Other Strengths and Weaknesses"

**Theoretical Claims:**

I checked the provided calculation steps in Appendices A.1 and A.2.

---

> ### Author Rebuttal · Authors · 2025-04-01
>
> 1. We would like to thank the reviewer for the helpful comments and suggestions. The updated figures and proof can be seen here: https://anonymous.4open.science/r/rebuttal_materials-14918/
>
> 2. Indeed, we didn’t notice the paper [R1]: "Another Look at Residual Dynamic Mode Decomposition in the Regime of Fewer Snapshots than Dictionary Size" and we would like to thank the reviewer for pointing it out. As the reviewer mentions, we indeed conducted our work before the emergence of [R1]. However, after reading this paper, we confirm that Algorithms 7 and 8 in [R1] are different from our proposed method. We first briefly go over both algorithms here: Algorithm 7 is almost the same as the kernel-ResDMD algorithm in the original ResDMD paper(Colbrook & Townsend (2024)) except for the condition of fewer snapshots than dictionary size. It still computes the $\epsilon$-pseudospectrum and corresponding approximate eigenfunctions as in ResDMD. The grid points $z_j$ serve as a candidate of eigenvalue to be tested on in the 3rd step of Algorithm 7, and will be kept only if its ‘’residual” is smaller than a threshold. However, the dictionary used in ‘’residual” is important. In our ResKoopNet method, the dictionary is obtained by minimizing ‘’spectral residual”, which is theoretically guaranteed by Appendix B.2 in the original ResDMD paper(Colbrook & Townsend (2024)). As for Algorithm 8, it is just a variant of Algorithm 7 where a specific candidate of eigenvalue $\lambda$ replaces the grid points $z_j$. So, we would like to point out that our contribution has nothing to do with any modification of improvement on any specific part in ResDMD such as pseudospectrum, rather we proposed a computing method that exploits the theoretical framework of spectral residual and addresses the  **spectral inclusion** problem, and validate it in a few examples including a real-world problem.
>
> 3. Based on the reviewer’s suggestion, we have improved the convergence analysis, which is mentioned in the file **convergence_proof.pdf**.  Since the function space $\mathcal{F}$ we considered here is $L^2$, we will be able to apply Theorem A.3. Under a few assumptions as in Assumption A.4, we have shown the convergence of the trained dictionary to the optimal dictionary and the convergence of estimated eigenpairs to the true spectrum of the Koopman operator $\mathcal{K}$. Now it not only shows the existence of the dictionary but also shows the convergence of the dictionary and we would like to thank the reviewer for the suggestion.
>
> 4. The last part of Section 3.2 named ‘’Computing Algorithm” on page 4 briefly introduces the architectures and layers, etc. We have investigated other parameter settings and tried different layer numbers. Specifically, in the pendulum example, we have scanned the layer number in the range [1,2,3,4] and layer size in the range of [250, 275, 300, 325, 350] and demonstrated the robustness of our parameter choices. This result is shown in **Fig_8.png**. The hyperparameter scanning results for the neural experiment are included in **Fig_17.png**: with smaller layer sizes and layer numbers, the clustering performance is not stable but becomes robust after the two hyperparameters reach a threshold. However, integrating the spectral residual-based Koopman operator approximation method with more complicated neural networks like PINO or KAN is beyond the scope of current work and will be addressed in future extensions.

---

### Official Review · Reviewer_1CwD · 2025-03-17

**Overall Recommendation:** 4

**Summary:**

The paper introduces ResKoopNet, a neural network-based method for learning Koopman operator representations of high-dimensional nonlinear dynamical systems.
ResKoopNet aims to address limitations of previous methods of learning Koopman operators from data such as Extended Dynamic Mode Decomposition that discover spurious eigenpairs from data, and the spectral inclusion problem related to the difficulty in capturing the entire true spectrum of the Koopman operator, especially for systems with continuous spectra.

The method extends Residual Dynamic Mode Decomposition (ResDMD) by explicitly minimizing a spectral residual loss function. The minimisation of the spectral residual contributes in avoiding spectral pollution, where the discretisation of the infinite dimensional operator to a finite matrix results in the discovery of spurious eigenvalues that are numerical artifacts.

The main contributions  of  this submission is using a feedforward neural network to automatically select the dictionary functions, overcoming the limitations imposed by predefined basis dictionaries used in the approximation of the operator, and using the spectral residual loss in the optimisation.
 Through minimising the spectral residual, ResKoopNet aims to approximate both discrete and continuous spectra.
Numerical results demonstrate ResKoopNet's accuracy on several example systems: on a classical pendulum system, on a turbulent flow, and on neural dynamics from mouse visual cortex.

For the pendulum system the proposed approach is shown to outperform existing methods in approximating the Koopman operator in terms of the number of basis functions (observables) required for accurate spectrum approximation (proposed method seems to require much fewer basis functions compared to ResDMD for same amount of data ) .

**Claims And Evidence:**

The authors claim that ResKoopNet addresses the spectral inclusion and spectral pollution issues of previous Koopman approximation methods. For the pendulum system the authors demonstrate that the method effectively tackles the spectral pollution issue and the spectral inclusion by increasing the number of observables.

**Essential References Not Discussed:**

I am not aware of any, but I do not word directly in this sub-field.

**Experimental Designs Or Analyses:**

Yes I checked the presented experiences and I have included some of my issues and comments also in the other comments

Regarding the experiment with the application on neural data:
- I am not sure why the authors choose to compare their method and the competing method in Figure 6A using a different number of eigenfunction (500 vs 50). In A9 the authors justify partly their choice, but I would expect to see how their method performs with 50 basis functions to compare it to Hankel-DMD, and to see in the main text a comparison with a competing methods with the same number of functions. Since the constraint for the Hankel-DMD and the kernel ResDMD is the number of snapshots I would expect first to see a comparison of all methods with this number of bases, and then additionally present results with more bases where possible. Thus, I am not entirely convinced about the validity of the remaining analyses performed on this dataset, but I remain open to being persuaded otherwise by the authors.

**Methods And Evaluation Criteria:**

- The use of spectral residual as a loss function is conceptually valid and justified given existing literature, making it suitable for evaluating the accuracy of spectral approximation of the operator.



- The choice of the pendulum system as a benchmark makes absolutely sense, because the method can be compared against ground truth. However the same is not true for the other two benchmarks. For the turbulent flow experiment I am not entirely convinced that the method outperforms competing methods (see my questions below), while regarding the neural data experiment I am not sure about the motivation of applying the method in this setting.

- However, more explicit justification for choosing specific hyperparameters (e.g. the size and structure of neural networks, number of eigenfunctions, and dimensionality reductions) would be beneficial. See also my questions below.

**Other Comments Or Suggestions:**

- Page 5: the authors write " Even with a much larger amount of dataset": I think they want to write amount of datapoints"
- the fonts in most figures are too small to be readable.
- In Fig. 5, since you are trying to relate the 2D pressure field of the original system to the first eigenfunction of the learned Koopman operator, I would suggest to rescale the colormaps so that the associated regions in the two upper plots have same/similar colouring. (minor)

**Other Strengths And Weaknesses:**

**Strengths:**
- Identifies Koopman operator by directly optimising the error of the approximation of the operator in the eigenspace
- Can approximate the operator for systems with continuous spectra, unlike EDMD-like approaches



**Weaknesses:**

- As mentioned by the authors the method is sensitive to hyperparameter selection.

- The writing and organising of the text could benefit from a bit of restructuring. For instance, the reference to the pseudospectrum in page 4 should be in my view under a separate subsection. Also when describing the experiments, the reader needs sometimes more background details to understand the setting. E.g. when describing the turbulent example the reader has to go to the paper of  Colbrook &
Townsend (2024) to understand the experimental setting.
- High computational demands discussed in A.5, but it is acceptable as first implementation of the approach that can be improved on in follow-up work

**Questions For Authors:**

- For the high dimensional turbulent system, do you reduce first the dimensionality of the state with SVD, and then compute the Koopman approximation? If yes, how do you select the reduced dimensionality? Can you show a plot with the singular values? Do you follow the same procedure for all methods?

- For the turbulent flow experiment, in the paper of Colbrook &Townsend (2024), the authors show both for DMD and ResDMD that the eigenfunctions have some characteristic details indicative of the acoustic sources. In the plots you provide in the main text from the proposed method, and in the appendix from the Hankel-DMD these details are absent. As a non-specialist, I wonder why this is the case, and whether you can still claim successful approximation of the operator, when such details are not captured by the proposed method.
- Related to the previous question, throughout the high dimensional experiments the authors seem to select to first reduce the dimensionality of the data to 300 through SVD and then apply the method. How do you select that value?

**Relation To Broader Scientific Literature:**

The method builds upon extensive literature on data-driven approximation of dynamical systems, and in particular on data-driven approximation of the Koopman operator. Specifically it extends recent work that uses the spectral residual to clear the identified spectral eigenpairs from spurious eigenpairs introduced through the numerical approximation of the infinite operator. It extends the existing literature by first using a neural network as a dictionary of observables/basis functions instead of using pre-selected basis, and uses the spectral residual as a loss function. Existing DMD methods optimise the dictionary of active basis function in the approximation by including a L1 cost on the basis coefficients.

**Theoretical Claims:**

I checked the derivations in A.2 and A.3

---

> ### Author Rebuttal · Authors · 2025-04-01
>
> 1. We would like to thank the reviewer for the helpful comments. The updated figures and proof can be seen here: https://anonymous.4open.science/r/rebuttal_materials-14918/
>
> 2. Regarding the benchmarking question:
>
> (1) In the 2nd experiment, the 300 basis functions we have chosen are indeed different from the 250 basis functions used in the original ResDMD paper (Colbrook & Townsend, 2024). We have re-done the experiment and the results are shown in **Fig5_new.png**;
>
> (2) In the neural experiment, to fairly compare with the 50 bases used in Hankel-DMD, we re-estimated the Koopman eigenfunctions using 50 dictionaries (24 SVD-truncated bases, one constant, and 25 trainable bases). The performance remained comparable to the case with 501 bases. See **Fig_6_new.png and Fig_13.png, Fig_16.png** for the averaged eigenfunctions and clustering results.
> We would like to clarify that the results are robust to hyperparameter choice (hidden layer number and neuron size in the hidden layer) when they reach a threshold for a given dataset. To illustrate this, we have added a hyperparameter scan result in **Fig_8.png** by scanning the hidden layer size from 1 to 4 and each layer’s neuron size in the range of [250, 275, 300, 325, 350]. The figure shows that the performance is robust with the increase of both hyperparameters, suggesting that varying the layer size will not trigger sensitive changes in the approximation results after a certain threshold. Similar robustness results are shown for the neural experiment in **Fig_17.png**: with smaller layer size and layer number, the clustering performance is not stable but becomes robust after the two hyperparameters reach a threshold. This justifies our network structure of 3 layers of 200 neurons.
>
> 3. We agree that the topic of “pseudospectrum” should be separated. We can put it at the bottom of Section 3.1 with the title “Continuous Spectra and Pseudospectrum”.
>
> 4. We will add some background details for turbulence example: ‘The turbulent flow dataset from Colbrook & Townsend (2024, Section 6.3) models a two-dimensional airfoil system with Reynolds number $3.88 \times 10^5$ and Mach number $0.07$. The data captures a pressure field at 295,122 spatial points across 798 time steps, sampled every $2 \times 10^{-5}$ seconds’.
>
> 5. We will extend the computational cost analysis and incorporate computational bottlenecks and theoretical extensions.
>
> 6. The colormap values of all Koopman modes in **Fig5_new.png** now have the same scale.
>
> 7. In the turbulence experiment, we will add further explanation of how truncated SVD is applied before using ResKoopNet. We applied a change of basis method here; specifically, consider the data matrix decomposed by truncated SVD $X=USV^\top$ with truncation $k=150$ singular values, then multiplying $V$ on both sides from the right to get $XV=US$, which is the lower dimensional data matrix projected by matrix $V$; then we apply ResKoopNet and compute its (low-dimensional) Koopman modes; then multiply the matrix $V^\top$ from the left to recover the original Koopman modes. Koopman modes and eigenfunctions are ranked by ascending spectral residuals, where smaller residuals indicate better approximations.
>
> 8. We added singular value plots in Figures 5(e) and (f) (see **Fig5_new.png**), revealing a significantly large singular value corresponding to the dominant spatial pattern captured by Koopman mode 1 in Figure 5(b).
>
> 9. Yes, we used SVD for both high-dimensional experiments(2nd and 3rd). In the turbulence example, we chose the 150 reduced dimension following Colbrook & Townsend, 2024. In the neural example, we reduce the data into 300 dimensions as it is a manageable dimension compared to the original dimension (>7000), but thanks to the reviewer we have demonstrated that with a smaller truncated dimension (24) we also get meaningful eigenfunctions (see previous reply). We also use the same method (filtered by spectral residual) in selecting eigenvalues in 1st pendulum experiment, selecting Koopman modes in 2nd turbulence experiment, and selecting eigenfunctions in 3rd neural experiment.
>
> 10. In **Fig5_new.png (c)(d)**, we have also illustrated the acoustic vibration and turbulent fluctuation characteristics. The original Hankel-DMD results in Appendix Figure 7 are similar to Figure 5(c)(d) and those in the ResDMD paper, which has no new characteristic to specify.
>
> 11. Regarding the SVD procedure: we first reduced spatial dimension to 150 using truncated SVD. Then, we applied ResKoopNet to obtain 1+150+149=300 Koopman modes ranked by ascending spectral residual. Here, '1' denotes a constant non-trainable basis, '150' are SVD-reduced spatial coordinates, and '149' are trainable bases (later adjusted to '99' for a total of 250 to match the original ResDMD setup). We then selected the less "polluted" (low-dimensional) Koopman modes and mapped them back to the original high-dimensional space using the SVD matrix $V$.

---

> > ### Comment · Reviewer_1CwD · 2025-04-08
> >
> > I thank the authors for their detailed reply to my comments (and to the comments of the other reviewers). Their responses have sufficiently addressed my concerns and I appreciate the additional effort they put in to clarify our concerns and improve their work. Therefore I will update my evaluation accordingly.
> >
> > However, I expect the authors to incorporate the suggested edits/clarifications that resulted from the reviews into their manuscript/supplement, since these details clarify their work and improve reproducibility.

---

> > > ### Author Response · Authors · 2025-04-08
> > >
> > > Thank you very much for your kind feedback and suggestion. I truly appreciate your time and comments, which have helped improve the clarity and quality of the work. While ICML did not allow manuscript updates during the rebuttal phase, I have incorporated all suggested edits and clarification into the updated version of the paper to ensure better reproducibility.

---

### Decision · Program_Chairs · 2025-05-01

**Decision:**

Accept (poster)

**Comment:**

This paper introduces ResKoopNet, a neural network-based framework for approximating the Koopman operator using a spectral residual loss. The method aims to address spectral pollution and inclusion issues prevalent in prior approaches such as EDMD and ResDMD. While the core idea of integrating neural dictionary learning with residual-based spectral refinement is incremental, the paper provides a thoughtful and well-motivated extension to existing Koopman operator methods. The experimental validation across diverse systems—including a pendulum, turbulent flow, and neural recordings—demonstrates the potential of the approach, although some concerns remain regarding fair comparisons (e.g., dictionary sizes across methods) and clarity in defining optimality and convergence. The authors were responsive during the rebuttal, conducting additional experiments and clarifying technical points, including convergence arguments and hyperparameter robustness. While the methodological novelty is modest, the empirical value, clarity of exposition, and relevance to ongoing work in Koopman learning justify a weak accept.